# Cytotoxic CD4+ T cells driven by T-cell intrinsic IL-18R/MyD88 signaling predominantly infiltrate *Trypanosoma cruzi*-infected hearts

Carlos-Henrique D Barbosa[1†‡], Fábio B Canto[1†§], Ariel Gomes[1†#], Layza M Brandao[1], Jéssica R Lima[1], Guilherme A Melo[1], Alessandra Granato[1¶], Eula GA Neves[2], Walderez O Dutra[2], Ana-Carolina Oliveira[3], Alberto Nóbrega[1], Maria Bellio[1]*

[1]Department of Immunology, Instituto de Microbiologia Paulo de Góes, Universidade Federal do Rio de Janeiro (UFRJ), Rio de Janeiro, Brazil; [2]Laboratório de Biologia das Interações Celulares, Universidade Federal de Minas Gerais, Belo Horizonte, Brazil; [3]Instituto de Biofísica Carlos Chagas Filho, Universidade Federal do Rio de Janeiro (UFRJ), Rio de Janeiro, Brazil

*For correspondence:
mariabellioufrj@gmail.com

[†]These authors contributed equally to this work

Present address: [‡]Department of Microbiology & Immunology, Chicago Medical School, Rosalind Franklin University of Medicine and Science, Chicago, IL, United States; [§]Departamento de Imunobiologia, Instituto de Biologia, Universidade Federal Fluminense (UFF), Niterói, Brazil; [#]Faculdade de Medicina da Universidade Estácio de Sá, Angra dos Reis, RJ, Brazil; [¶]The Hospital for Sick Children - Genetics and Genome Biology, Toronto, ON, Canada

**Competing interest:** The authors declare that no competing interests exist.

**Abstract:** Increasing attention has been directed to cytotoxic CD4+ T cells (CD4CTLs) in different pathologies, both in humans and mice. The impact of CD4CTLs in immunity and the mechanisms controlling their generation, however, remain poorly understood. Here, we show that CD4CTLs abundantly differentiate during mouse infection with the intracellular parasite *Trypanosoma cruzi*. CD4CTLs display parallel kinetics to Th1 cells in the spleen, mediate specific cytotoxicity against cells presenting pathogen-derived antigens and express immunoregulatory and/or exhaustion markers. We demonstrate that CD4CTL absolute numbers and activity are severely reduced in both *Myd88*-/- and *Il18ra*-/- mice. Of note, the infection of mixed-bone marrow chimeras revealed that wild-type (WT) but not *Myd88*-/- cells transcribe the CD4CTL gene signature and that *Il18ra*-/- and *Myd88*-/- CD4+ T cells phenocopy each other. Moreover, adoptive transfer of WT CD4+GzB+ T cells to infected *Il18ra*-/- mice extended their survival. Importantly, cells expressing the CD4CTL phenotype predominate among CD4+ T cells infiltrating the infected mouse cardiac tissue and are increased in the blood of Chagas patients, in which the frequency of CD4CTLs correlates with the severity of cardiomyopathy. Our findings describe CD4CTLs as a major player in immunity to a relevant human pathogen and disclose T-cell intrinsic IL-18R/MyD88 signaling as a key pathway controlling the magnitude of the CD4CTL response.

## Editor's evaluation

This is an interesting study, conducted in mice, that demonstrates for the first time the presence of a large population of cytotoxic CD4+ T lymphocytes in infection with Trypanosoma cruzi, a relevant human pathogen. At present, the relevance of these cells in protective immunity engendered by the host remains unclear. Additional experiments are needed to characterize the functionality of these cytotoxic CD4 T cells vis-a-vis the canonical Th1 T cells. This paper can be of interest to scientists interested in immune responses to parasitic infections.

## Introduction

Cytotoxic CD4$^+$ T cells (CD4CTLs) are present in viral infections, cancer and autoimmune diseases (*Cheroutre and Husain, 2013*; *Juno et al., 2017*; *Takeuchi and Saito, 2017*). Although more frequently found in chronic conditions, CD4CTLs have also been described during acute viral infection (*Brien et al., 2008*; *Brown et al., 2012*; *Fang et al., 2012*; *Jellison et al., 2005*; *Wilkinson et al., 2012*). At present, however, there are few studies on the CD4CTL response to infection with non-viral intracellular pathogens (*Burel et al., 2016*; *Canaday et al., 2001*; *Kotov et al., 2018*; *Krueger et al., 2021*). The role of CD4CTLs in immunity against intracellular parasites and the mechanisms operating in the generation and maintenance of this cell subset are poorly understood.

Following the recognition of the cognate antigen-MHC complex and under the influence of cytokines secreted by antigen presenting cells (APCs), naïve CD4$^+$ T cells proliferate and differentiate into diverse subsets, including Tfh, Th1, Th2, or Th17, with varied effector functions. Different pathogen-derived molecular patterns (PAMPs) elicit distinct innate immune pathways, resulting in tailored Th responses against the infectious agent (*Kara et al., 2014*). Interestingly, single-cell heterogeneity in the transcriptional profile of effector CD4$^+$ T cells, dictated in a microbe-specific manner, has been recently found in the gut (*Kiner et al., 2021*). Additionally, tissue-derived factors can also be involved in defining CD4$^+$ T cell phenotypes (*Poholek, 2021*). In this context, the mechanism regulating the differentiation of CD4CTLs remains an open and relevant question. Although some authors consider CD4CTLs functional variants of Th1 cells, there is also evidence that CD4CTLs might constitute a Th1-independent subset, directly differentiating from naïve CD4$^+$ T cell precursors or even a heterogeneous population congregating cytolytic lymphocytes that originate from different T-cell subsets, such as Treg, Th1, and Th2 cells (reviewed in *Cheroutre and Husain, 2013* and *Takeuchi and Saito, 2017*).

Receptors belonging to the TLR and IL-1R families are upstream of the MyD88 adaptor molecule and are expressed both in myeloid and lymphoid cell lineages. MyD88-dependent signaling in innate cells can influence Th1 cell differentiation by inducing the release of IL-12 or IFN-γ (*Joffre et al., 2009*; *Takeda et al., 1998*). The T-cell intrinsic MyD88 adaptor function was also shown to be essential for the Th1 response in different contexts. The identity of the receptor acting upstream of MyD88 on T cells, however, was not always determined (*Frazer et al., 2013*; *LaRosa et al., 2008*) and remains a matter of debate. Early in vitro studies have shown that IL-18 synergizes with IL-12 in the induction of IFN-γ production and T-bet expression by CD4$^+$ T cells and, therefore, impacts Th1 development (*Robinson et al., 1997*). Intriguingly, the IL-18 contribution for a robust Th1 response in vivo has not always been confirmed (*Haring and Harty, 2009*; *Monteforte et al., 2000*). ST2 (IL-33 receptor) and IL-1R have also been implicated in the establishment of Th1 responses to viral infection (*Baumann et al., 2015*) and immunization (*Ben-Sasson et al., 2009*; *Schenten et al., 2014*), respectively. Previously, our group identified the crucial role of T cell-intrinsic IL-18R/MyD88 signaling for the reinforcement and maintenance of the Th1 program in vivo, in response to mouse infection with *Trypanosoma cruzi*, in which Th1 cells have a key protective function (*Oliveira et al., 2017*; *Rodrigues et al., 2012*).

*T. cruzi* is an intracellular protozoan parasite and the etiologic agent of Chagas disease (*American trypanosomiasis*), one of the most important parasitic diseases in the Western Hemisphere. Chagas disease is an important cause of heart failure, stroke, arrhythmia, and sudden death, being the leading cause of cardiomyopathy among younger individuals in Latin America (*Bern, 2015*; *Nunes et al., 2018*). Here we showed that CD4$^+$ T lymphocytes with a CTL phenotype are abundantly present in the spleen of mice infected with *T. cruzi* and display in vitro and in vivo Ag-specific cytotoxicity against APCs. Of note, we found that cells with a CTL phenotype predominate among CD4$^+$ T lymphocytes infiltrating infected cardiac tissue. Using mixed bone marrow (mix-BM) chimeras, we show that T cell-intrinsic IL-18R/MyD88 signaling plays a crucial role for the expansion/survival of CD4$^+$ T cells expressing the CTL gene signature in vivo. In accordance with results obtained in mix-BM chimeras, we found that absolute numbers of CD4 T cells displaying cytotoxic markers and activity were severely reduced in *Myd88$^{-/-}$* and *Il18ra$^{-/-}$* mouse lineages. Moreover, adoptive transfer of wild-type (WT) CD4$^+$GzB$^+$ T cells to *Il18ra$^{-/-}$* mice increase survival of this susceptible mouse strain to infection. Interestingly, the majority of CD4$^+$GzB$^+$ T cells in WT infected mice express high levels of immunoregulatory molecules such as, Tim-3, Lag-3, CD39, and PD-1. Finally, and importantly, we detected the presence of CD4$^+$ T cells expressing perforin in the blood of patients with the severe cardiac form of Chagas disease and correlation analyses point to a possible pathogenic role of these cells in the chronic phase of Chagas myocarditis. To our knowledge, the present study is the first to extensively characterize CD4CTLs

generated in response to a protozoan pathogen in a murine model and to elucidate a central mechanism that accounts for their expansion/survival.

## Results

### CD4$^+$ T cells with cytolytic phenotype are abundantly present in the spleen of mice infected with *T. cruzi*

CD4CTLs express granzyme B (GzB) and perforin (PRF), which are major constituents of CD8CTL and NK cell-granules that promote target cell death in a number of ways (*Afonina et al., 2010*). Here, we reveal the presence of a TCRαβ$^+$CD8α$^-$NK1.1$^-$CD4$^+$GzB$^+$ T cell population, which represents around 30% of the CD4$^+$ T cells in the spleen of C57BL/6 (B6) mice, in the acute phase of infection with *T. cruzi* (*Figure 1A and B* and *Figure 1—figure supplement 1A*). Equivalent percentages of CD4$^+$PRF$^+$ T cells were found (*Figure 1C*) and the majority of CD4$^+$GzB$^+$ T cells express PRF and the cytolytic marker 2B4 (CD244) (*Figure 1D and E*). Of note, most CD4$^+$GzB$^+$ T cells (>80%) do not express IFN-γ at day 14 post-infection (pi), the peak time-point of the T cell response, and although CD4GzB$^+$ and CD4IFN-γ$^+$ cells follow similar kinetics in the spleen, the absolute number of CD4$^+$GzB$^+$ T cells was superior to that of CD4$^+$IFN-γ$^+$ cells, at every time point of the infection (*Figure 1F and G*). It is known that intracellular parasitism by *T. cruzi* Y strain in the spleen is at its highest point between day 12 and 15 pi, concomitant with the histologically observed massive destruction of parasitized splenic macrophages (*Cordeiro et al., 1997*). Therefore, there is a coincidence between parasite titers in the spleen and the frequency and absolute numbers of CD4$^+$GzB$^+$ T cells. We also observed an increase in the percentage of degranulating CD4$^+$ T cells at day 14 pi, comparable to the CD8$^+$ T cells, as inferred by staining with anti-CD107a (LAMP-1) mAb, (*Figure 1H*), although CD8CTLs are present at higher percentage (*Figure 1—figure supplement 1B-E and G* ) and absolute number (*Figure 1—figure supplement 1F and H*). In summary, these data show the remarkable induction of CD4$^+$ T cells with cytolytic potential during infection with *T. cruzi*, an effector subpopulation whose appearance in the spleen parallels and largely outnumbers that of classic GzB$^-$ IFN-γ$^+$ Th1 cells.

### CD4$^+$ T cells from infected spleen exert Ag-specific cytotoxicity

To ascertain the presence of CD4$^+$ T lymphocytes with cytolytic function, we performed a redirected cytotoxicity assay, with anti-CD3 mAb–coated B cell blasts as target cells that were co-incubated with either highly purified effector (CD44$^{hi}$) CD4 T cells sorted from the spleen of infected mice or naive (CD44$^{lo}$) CD4 T cells from non-infected controls. The CTL activity was observed only when effector CD4$^+$ T cells were employed (*Figure 2A*). In order to determine if the CD4CTL killing capability is dependent on GzB- and PRF-mediated killing, we performed the cytotoxic assays in the presence of the GzB and PRF inhibitors, Z-AAD-CMK and Concamycin A, respectively. As shown in *Figure 2—figure supplement 1A*, the killing of target cells was abolished in this condition, demonstrating that the cytolysis mediated by CD4CTLs in *T. cruzi*-infected mice depends on GzB and PRF, as previously shown for tumor-specific CD4CTLs (*Quezada et al., 2010*). Antigen-specific cytotoxicity, using highly purified CD4$^+$ T cells, was also assayed in vitro against an I-A$^{b+}$ macrophage cell line (IC-21), loaded or not with *T. cruzi* amastigote total protein extract. Results shown in *Figure 2B* and *Figure 2—figure supplement 1B* demonstrate that CD4CTLs with cognate cytotoxic activity are present in the spleen of infected mice. FasL is expressed on CD8$^+$ and CD4$^+$ T cells at the immunological synapse following Ag-triggered degranulation and participates in CD4CTL-mediated lysis of target cells (*Bossi and Griffiths, 1999*; *Brown et al., 2009*; *Kotov et al., 2018*). As shown in *Figure 2B*, addition of anti-FasL mAb significantly inhibited the specific lysis of the target cells, demonstrating the involvement of the FasL/Fas pathway in the killing process.

We tested the in vivo killing of target splenocytes loaded with class II-restricted peptides (SA85-1.1c and d), derived from a protein of the *T. cruzi* transialidase superfamily (*Millar and Kahn, 2000*; *Figure 2C* and *Figure 2—figure supplement 1C*). The detailed Ag-specific repertoire of CD4CTLs could not be further assessed because of the scarce definition of parasite-derived CD4 epitopes in the literature. To circumvent this, we infected mice with ovalbumin (OVA)-expressing Y strain trypomastigotes (Y-OVA) (*Gomes-Neto et al., 2018*) and compared the in vivo cytotoxic response against known OVA-derived CD4 and CD8 epitopes: OVA$_{323-339}$ and OVA$_{257-264}$, respectively. As shown in *Figure 2D*, the same level of specific killing was attained against targets loaded with class I- or class II-restricted

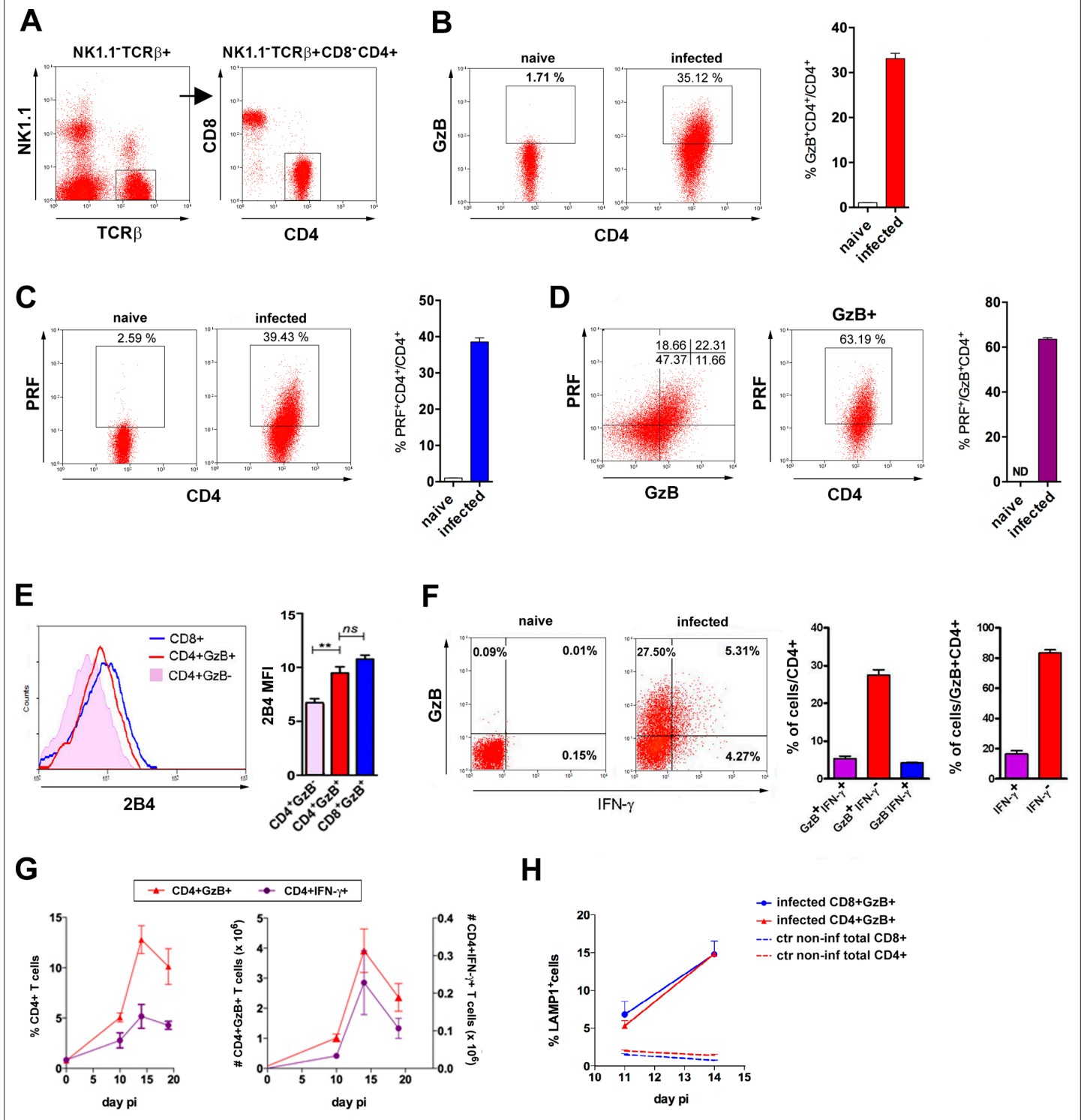

**Figure 1.** CD4+GzB+PRF+ T cells are present at high frequency in the spleen of mice infected with *T. cruzi*. (**A**) CD4+ T cell gating strategy. (**B and C**) Representative flow-cytometry dot plots of granzyme B (GzB) (**B**) or perforin (PRF) (**C**) staining, gated on CD4+ T cells from non-infected (naïve) or infected B6 mice (left) and mean frequency of CD4+GzB+ (or CD4+PRF+) T cells in the spleen (right). (**D**) Representative dot plots of PRF and GzB expression in CD4+ T cells gated as in (**A**) and mean frequencies of PFR+ among GzB+ CD4+ T cells (isotype controls on *Figure 1—figure supplement 1A*). (**E**) Representative histogram of 2B4 (CD244) expression on CD8+ (blue), CD4+GzB+ (red) or CD4+GzB- T cells (pink), and correspondent mean fluorescence intensity (mean MFI) values; (**F**) Representative dot plots of GzB and IFN-γ staining in gated CD4+ T splenocytes (left), mean frequencies of GzB+IFN-γ- (red), GzB+IFN-γ+ (purple) and GzB-IFN-γ+ (blue) subsets among total CD4+ (center) or gated on CD4+GzB+ T cells (right). (**G**) Kinetics of the mean frequency of CD4+IFN-γ+ (purple) and CD4+GzB+ (red; left), and corresponding absolute numbers (right). (**H**) Frequency of LAMP-1+(CD107a+) cells

*Figure 1 continued on next page*

*Figure 1 continued*

among CD4⁺GzB⁺ (red) or CD8⁺GzB⁺ (blue) T cells in the infected spleen. Data obtained from individually analyzed mice (n=4), at indicated or 14 day post-infection (pi). Error bars = SEM. ND = non-detected. ns = non-significant; **p≤0.01 (Student's t-test). Data are representative of three or more independent experiments.

The online version of this article includes the following figure supplement(s) for figure 1:

**Figure supplement 1.** Comparative percentages and absolute numbers of granzyme B-positive (GzB⁺) and perforin-positive (PRF⁺) T cell subsets.

OVA peptides. Together, these experiments confirmed the presence of a CD4⁺ T cell population with Ag-specific cytolytic function against MHC class II-restricted epitopes in the spleen of mice infected with *T. cruzi*.

## CD4⁺GzB⁺ T cells express transcription factors typical of CTLs

We then investigated the expression of transcription factors (TFs) typical of cytotoxic CD8 T cells. T-bet and Eomesodermin (Eomes) are known to bind to the promoters of *Gzmb* and *Prf1* (*Glimcher et al., 2004*; *Intlekofer et al., 2005*; *Pearce et al., 2003*) and were shown to be expressed in CD4CTLs (*Curran et al., 2013*; *Eshima et al., 2012*; *Qui et al., 2011*). *Figure 2E and F* show that the vast majority of CD4⁺GzB⁺ T cells express T-bet, but not FoxP3, the master gene of Treg cells. In accordance with the cytolytic potential of CD4⁺GzB⁺ T cells in our model, 60% of the CD4⁺GzB⁺ T cells express Eomes, while only a small percentage of GzB⁻ CD4⁺ T cells are Eomes⁺ (*Figure 2G*). Of note, at day 14 pi, the vast majority of both GzB⁺ and GzB⁻ CD4⁺ T lymphocytes are CD44^hi CD62L^lo effector/memory cells (*Figure 2—figure supplement 2*).

CD4CTLs that migrate to the gut, gradually stop expressing ThPOK, while upregulating the expression of RUNX3 (*Mucida et al., 2013*; *Reis et al., 2013*), which is known to drive the CTL gene program in CD8⁺ T cells (*Cruz-Guilloty et al., 2009*; *Egawa et al., 2007*). The long-form transcript encoding RUNX3 (*Runx3d*) is also expressed in Th1 cells, but not in Th2 or naïve cells (*Djuretic et al., 2007*). As shown in *Figure 2H*, besides *Runx3d* mRNA, we found that CD4⁺GzB⁺ T cells, sorted from the spleen of tamoxifen-treated infected GzmbCreER^T2/ROSA26EYFP reporter mice (*Bannard et al., 2009*; *Figure 2—figure supplement 3A*), express ThPOK (*Zbtb7b gene*)-coding RNA comparable to naive CD4⁺ T cells, at day 14 pi, when their number and frequency are at the highest levels in this organ (*Figure 1G*). The absence of CD8α-coding transcripts confirms the purity of sorted CD4⁺GzB⁺ T cells (*Figure 2—figure supplement 3B*). Since a large fraction of CD4⁺GzB⁺ T cells do not express EYFP in the infected reporter mice and, as a consequence, we could not sort bona-fide GzB⁻ CD4⁺ T cells, it was not possible to compare the levels of *Runx3d* expression between GzB⁺ and other subsets of activated CD4⁺ T cells. These data show that CD4⁺GzB⁺ T cells generated in response to infection with *T. cruzi*, besides expressing cytolytic markers, also express T-bet, Eomes and RUNX3, typical TFs found in cytotoxic T cells.

## CD4⁺GzB⁺ T cells express cytotoxic markers and immunoregulatory molecules

Tr1 cells are a subset of IL-10-producing CD4⁺FoxP3⁻ T cells that express GzB and PRF and exert a potent suppressive function both in humans and mice (*Roncarolo et al., 2018*). Since a high percentage of CD4⁺GzB⁺ T cells in our system express Eomes, which drives the development of Tr1 cells (*Zhang et al., 2017*), we investigated whether *T. cruzi*-induced CD4⁺GzB⁺ T cells would share other phenotypic characteristics with Tr1 cells. We detected a very low frequency of IL-10-producing TCRαβ⁺CD4⁺ T cells in the spleens of mice infected with *T. cruzi*, in agreement with previous data (*Jankovic et al., 2007*), although IL-10⁺ cells were found to be more frequent among CD4⁺GzB⁺ T cells (*Figure 3A* and *Figure 3—figure supplement 1A, B*). The co-expression of Lag-3 and CD49b defines both human and mouse Tr1 cells (*Gagliani et al., 2013*). Although most splenic CD4⁺GzB⁺ T cells express Lag-3, only a minor percentage of these cells co-express CD49b (*Figure 3B and C* and *Figure 3—figure supplement 1C*). On the other hand, a substantial fraction of splenic CD4⁺GzB⁺ T cells express the immunoregulatory molecule Tim-3 and downregulate CD27 (*Figure 3D and E*), indicating that these might be terminally differentiated cells, as suggested in other models (*Anderson et al., 2016*; *Takeuchi and Saito, 2017*).

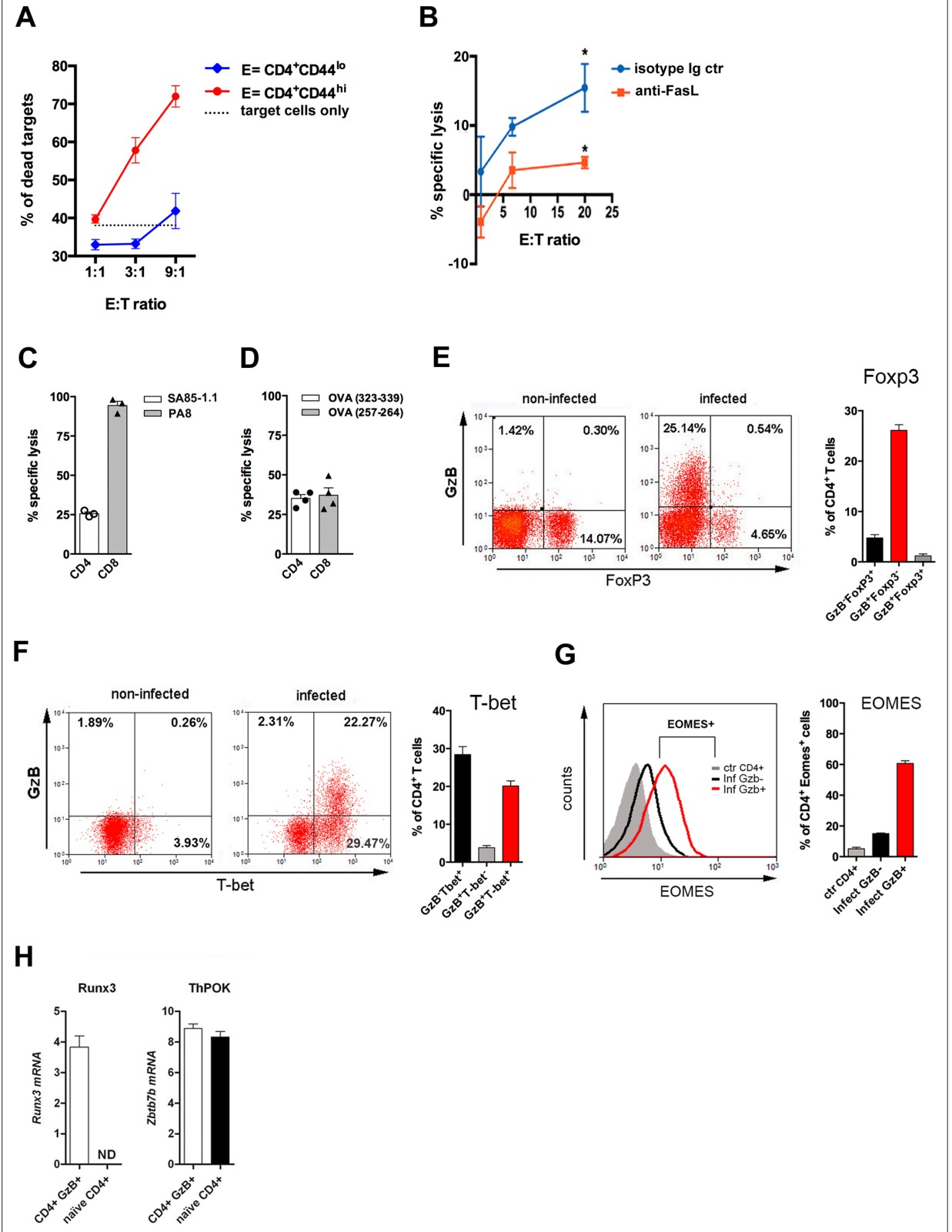

**Figure 2.** CD4 T cell-mediated Ag-specific cytotoxicity and transcription factor (TF) expression by CD4+GzB+ T cells. (**A**) Mean frequencies of dead target cells (LPS-induced B blasts) in redirected cytotoxicity assay employing sorted naïve (blue) or in vivo activated (red) CD4+ T cells as effectors, at different E:T ratios, in triplicates. (**B**) Specific lysis of IC-21 macrophages (T) loaded with amastigote antigens, co-incubated with highly purified effector CD4+ T cells, at the indicated E:T ratios, in the presence of anti-FasL mAb (red) or isotype control mAb (blue). Mean values of triplicates ± SEM are

*Figure 2 continued on next page*

*Figure 2 continued*

shown in (**A**) and (**B**); *p≤0.05 (two-tailed Student's t-test). Data are representative of three independent experiments. Gating strategy on ***Figure 2—figure supplement 1B***. (**C and D**) Percentages of Ag-specific cytotoxicity in vivo; bars represent mean values + SEM (error bars) of individually analyzed mice (n=3) infected with *T. cruzi* Y in (**C**) or ovalbumin (OVA)-expressing Y strain trypomastigotes (Y-OVA) strain in (**D**); class I-restricted (PA8 or OVA$_{257-264}$) and class II-restricted (SA851.1c+d or OVA$_{323-339}$) peptides were employed. Data are representative of two independent experiments. Cell survival was measured by flow cytometry (***Figure 2—figure supplement 1C***) and specific lysis calculated as described in Methods. (**E and F**) Representative dot plots of CD4$^+$ T cells (gated as in ***Figure 1A***) and corresponding mean frequencies of gated subpopulations: (**E**) granzyme B (GzB) and FoxP3 staining and (**F**) GzB and T-bet staining. (**G**) Representative histogram of Eomes expression in CD4$^+$ T splenocytes of non-infected mice (ctr, gray) and in GzB$^-$ (black) and GzB$^+$ (red) CD4$^+$ T cells from infected mice, gated as in ***Figure 1A and B*** (left) and mean frequencies of Eomes$^+$ cells among these different subpopulations (right). All experiments were done at day 14 post-infection (pi). Bars represent mean values of non-infected ctr group n=3 or infected group, n=3–5, individually analyzed mice. Error bars = SEM. (**H**) Runx3d and ThPOK (*Zbtb7b*) mRNA expression estimated by qRT-PCR in CD4$^+$EYFP$^+$ T cells sorted from infected and tamoxifen-treated GzmbCreER$^{T2}$/ROSA26EYFP mice, as described in Methods and shown in ***Figure 2—figure supplement 3A***; ND = not detected. Mean values of triplicates + SEM are shown. Data are representative of three independent experiments.

The online version of this article includes the following figure supplement(s) for figure 2:

**Figure supplement 1.** Cytotoxicity assays.

**Figure supplement 2.** The majority of CD4$^+$ T lymphocytes (GzB$^+$ or GzB$^-$) in the spleen of infected mice are non-naïve (activated effectors and memory) cells.

**Figure supplement 3.** Expression of *Tbx21* and *Cd8a* in sorted GzB$^+$CD4$^+$ T cells.

Natural killer group 2 (NKG2) members A, C, D, and E are C-type lectin-like receptors and cytotoxic markers expressed by NK, NKT, γδ, and activated CD8$^+$ T cells (***Gunturi et al., 2004***; ***Stojanovic et al., 2018***). NKG2C/E and NKG2D molecules identify a CD4CTL population, present in mouse lungs during influenza A virus (IAV) infection (***Marshall et al., 2017***). As shown in ***Figure 3F and G***, the fraction of cells expressing NKG2A/C/E and NKG2D was increased among splenic CD4$^+$GzB$^+$ T cells, when compared to either GzB$^-$ CD4$^+$ T cells of infected mice or to naive CD4$^+$ T cells of non-infected controls (***Figure 3—figure supplement 2A and B***). The class I-restricted T cell associated molecule (CRTAM) was described as being critical for the differentiation of CD4CTLs (***Takeuchi et al., 2016***; ***Takeuchi and Saito, 2017***). ***Figure 3H*** shows that the frequency of CRTAM-expressing cells was also increased among splenic CD4$^+$GzB$^+$ T cells of *T. cruzi*-infected mice, when compared with GzB$^-$ CD4$^+$ T cells. We then analyzed whether CRTAM and NKG2X cytotoxic markers were co-expressed on GzB$^+$CD4$^+$ T cells. As shown in ***Figure 3—figure supplement 3***, while most NKG2D$^+$ cells also express the CRTAM cytotoxic marker, only a minority of CD4$^+$GzB$^+$ T cells co-express CRTAM and NKG2A/C/E molecules. The frequencies of CRTAM$^+$, NKG2A/C/E$^+$ and NKG2D$^+$ CD8$^+$GzB$^+$ T cells in infected mice are shown in ***Figure 3—figure supplement 4***. Since, as shown in ***Figure 2—figure supplement 2***, both GzB$^-$ and GzB$^+$ CD4$^+$ T cells are in their majority effector/memory cells at day 14 pi, equivalent results were obtained gating on CD44$^{hi}$ cells or not (***Figure 3—figure supplement 5***). Together, these results show that a significant fraction of the CD4$^+$GzB$^+$ T cell subpopulation found in this unicellular parasite infection model complies with the general phenotype of CD4CTLs and CD4CTL-precursors reported in response to viral infection.

## CD4CTLs are present in lower numbers in *Myd88*$^{-/-}$ infected mice

To extend our understanding on the requirements for the differentiation of CD4CTLs and their relationship to Th1 cells, we next investigated the role of IL-12, IFN-γ and MyD88 in their generation. IL-12 and IFN-γ are the major cytokines driving Th1 differentiation and T-bet expression, but their requirement for CD4CTL generation is controversial (***Brown et al., 2009***; ***Cooper et al., 2004***; ***Curran et al., 2013***; ***Hua et al., 2013***; ***Kotov et al., 2018***; ***Krueger et al., 2021***; ***Tagawa et al., 2016***; ***Xie et al., 2010***). Here, we observed that *Il12p40*$^{-/-}$ mice infected with *T. cruzi* displayed lower percentages of T-bet$^+$CD4$^+$ and IFN-γ$^+$CD4$^+$ T cells and exhibited very low levels of CD4$^+$GzB$^+$ T cells (***Figure 4A***). Interestingly, in infected *Ifng*$^{-/-}$ mice, despite the lower frequency of T-bet$^+$CD4$^+$ T cells, CD4$^+$GzB$^+$ T cells attained the same frequencies as observed in the spleens of WT animals (***Figure 4B***). However, absolute numbers of both total CD4$^+$ and CD4$^+$GzB$^+$ T cells were lower in *Ifng*$^{-/-}$ mice when compared to WT controls. Thus, both IL-12p40 and IFN-γ play important but distinct roles in the differentiation and expansion/survival of CD4$^+$GzB$^+$ T cells.

The above results obtained in *Il12p40*$^{-/-}$ and *Ifng*$^{-/-}$ mice, together with the fact that *Myd88*$^{-/-}$ mice have lower levels of Th1 cells and are highly susceptible to infection with *T. cruzi* (***Campos et al.,***

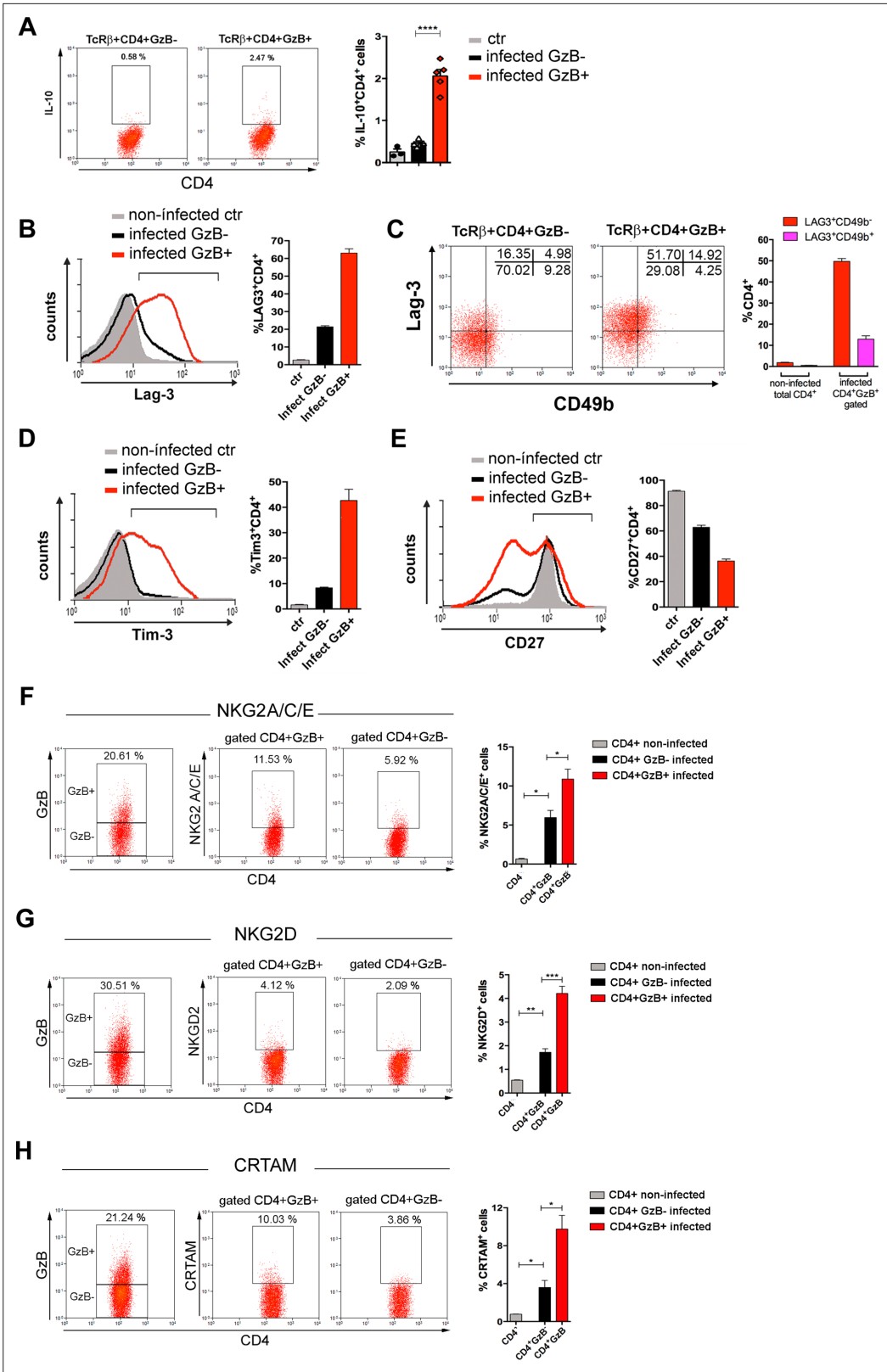

**Figure 3.** Expression of IL-10, immunoregulatory molecules and cytotoxic markers by CD4⁺GzB⁺ T cells. (**A**) Representative dot plots of IL-10 staining and mean frequencies of IL-10⁺ cells among non-infected (ctr, gray) or granzyme B-negative (GzB⁻, black) and GzB⁺ (red) CD4⁺ T cells from infected mice. (**B**) Representative histogram of Lag-3 staining and mean frequency of Lag-3⁺ cells. (**C**) Representative dot plots of Lag-3 and CD49b staining and

*Figure 3 continued on next page*

*Figure 3 continued*

mean frequency of different cells subsets. (**D**) Representative histogram of Tim-3 staining and mean frequency of Tim-3+ cells among different cells subsets. (**E**) Representative histogram of CD27 staining and mean frequency of CD27+ cells. (**A–E**) CD4+GzB+ T cells gated as in *Figure 1A, B*; staining controls in *Figure 3—figure supplement 1*. (**F–H**) Representative dot plots and mean frequencies of (**F**) NKG2A/C/E, (**G**) NKG2D, and (**H**) CRTAM expression among gated CD4+ T splenocytes. All experiments were done at day 14 post-infection (pi). Non-infected ctr group, n=3; infected group, n=3–5 mice, individually analyzed. Error bars = SEM; *p≤0.05; **p≤0.01; ***p≤0.001; ****p≤0.0001 (two-tailed Student's t-test). Data are representative of three independent experiments. Staining controls are shown on *Figure 3—figure supplement 2*.

The online version of this article includes the following figure supplement(s) for figure 3:

**Figure supplement 1.** IL-10, Lag-3 and CD49b expression.

**Figure supplement 2.** Representative dot plots and gating strategies for NKG2A/C/E, NKG2D and CRTAM staining on CD4+ and CD8+ GzB+ T cells.

**Figure supplement 3.** Most CD4+GzB+NKG2D+ T cells express the CRTAM cytotoxic marker while only a minority of CD4+GzB+ T cells co-expresses CRTAM and NKG2A/C/E molecules.

**Figure supplement 4.** Expression of NKG2A/C/E, NKG2D, and CRTAM cytotoxic markers by CD8+GzB+ T cells.

**Figure supplement 5.** Frequency of CRTAM-expressing cells among splenic granzyme B-positive (GzB+) and GzB- CD4+ T cells, gating or not on CD44hi cells.

*2004*; *Oliveira et al., 2010*), prompted us to address the role of MyD88-dependent pathways in the generation of CD4CTLs. Although infected *Myd88-/-* mice displayed equivalent absolute numbers and percentages of total splenic CD4+ T cells when compared to WT mice (*Figure 4—figure supplement 1A*), the frequencies and absolute numbers of CD4+GzB+ T cells were significantly lower in this mouse strain (*Figure 4C–E*). Furthermore, absolute numbers of CRTAM+, NKG2D+ and NKG2A/C/E+ CD4+GzB+ T cells were severely diminished in *Myd88-/-* mice (*Figure 4C–E*). This picture does not result from lower numbers of total activated CD4+ T cells in infected *Myd88-/-* mice, as shown by the percentages and absolute numbers of CD44hi cells, which are equivalent to those observed in WT animals (*Figure 4—figure supplement 2*). On the other hand, absolute numbers of CRTAM+, NKG2D+ and NKG2A/C/E+ CD8+GzB+ T cells were not affected in *Myd88-/-* mice (*Figure 4—figure supplement 1C-D*). Concerning the expression of perforin, both the percentages and absolute numbers of PRF+CD4+ T cells were also severely decreased in *Myd88-/-* compared to WT mice (*Figure 4F*). The same was observed when analyzing the frequency and absolute numbers of PRF+CD4+ T cells among GzB-expressing cells (*Figure 4G*). In order to establish to what extent the absence of MyD88 expression would impact on cytotoxicity mediated by CD4+ T splenocytes, we sorted activated CD4+CD44hi T cells from both WT and *Myd88-/-* infected mice to a high purity level and performed redirected cytotoxicity assay. Lower levels of cytotoxicity were observed when sorted *Myd88-/-* CD4+CD44hi T cells were employed, as compared to assays using equal numbers of sorted WT CD4+CD44hi T cells (*Figure 4H*). Together, these results demonstrate the important role of MyD88 signaling for the establishment of a robust cytotoxic response mediated by CD4CTLs.

## T cell-intrinsic IL-18R/MyD88 signaling plays a crucial role in the establishment of a CD4+ T cell subpopulation bearing the CD4CTL gene signature

MyD88 deficiency results in lower IL-12 production by APCs (*Campos et al., 2004*) and this could explain the diminished numbers of CD4CTLs found in infected *Myd88-/-* mice. However, we have recently shown that IL-18R/MyD88 signaling intrinsic to CD4+ T cells is critical for Th1 cell proliferation and resistance to apoptosis in response to infection with *T. cruzi* (*Oliveira et al., 2017*). If CD4CTLs herein described share a common early differentiation process with (or derive from) Th1 cells, it is reasonable to expect that besides IL-12 and IFN-γ, T-cell-intrinsic MyD88-dependent signaling would also be important for the generation of a robust CD4CTL response. In order to test this hypothesis, we constructed mix-BM chimeras in which WT (B6 × B6.SJL) F1 mice were irradiated and reconstituted with a 1:1 mix of WT and *Myd88-/-* BM cells. After reconstitution, mix-BM chimeras were infected with *T. cruzi* and, at day 14 pi, we sorted WT (CD45.1) and *Myd88-/-* (CD45.2) CD44+CD4+ T cells to a high purity level for transcriptome analysis. We then investigated whether

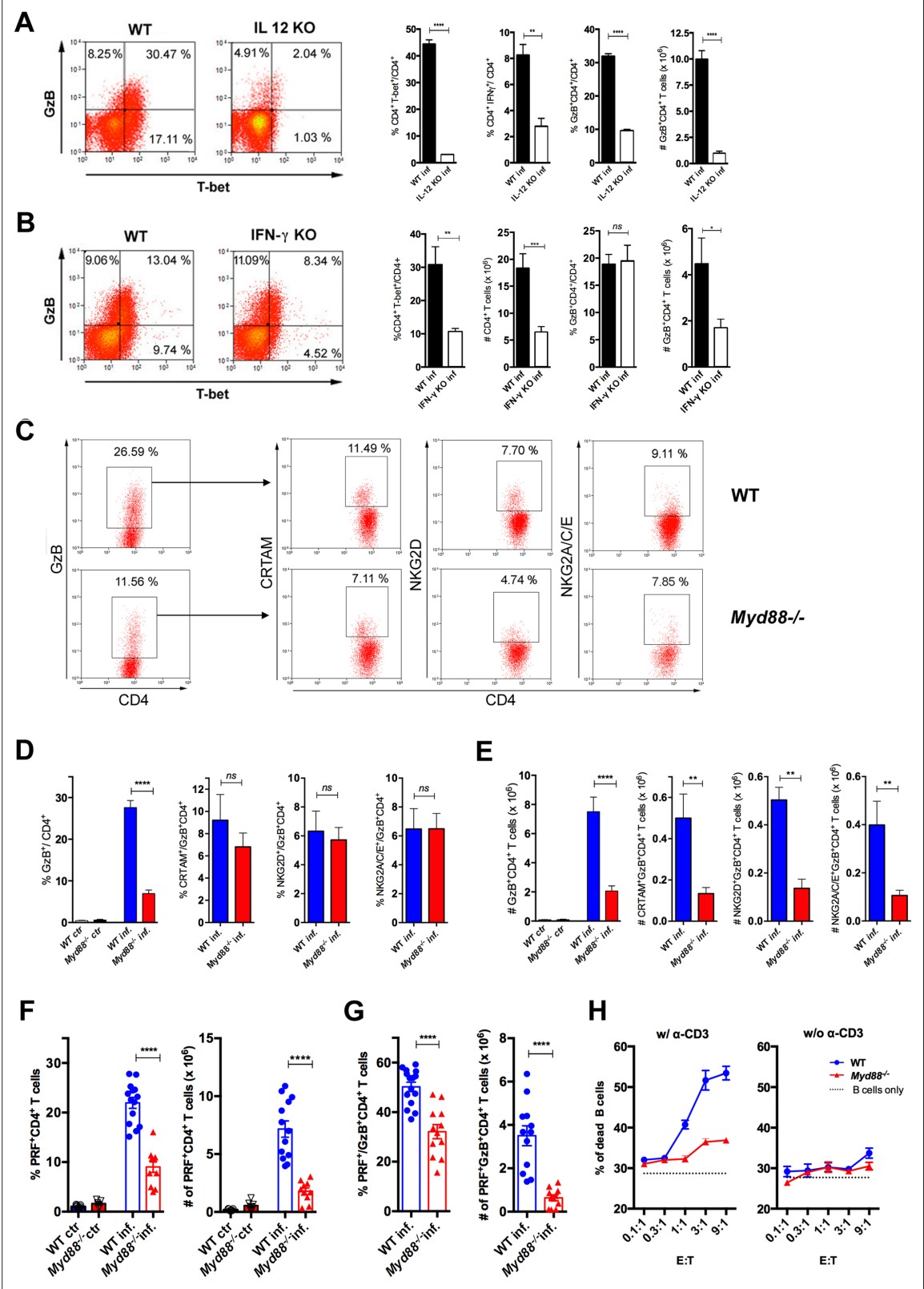

**Figure 4.** Cytotoxic CD4+ T cells (CD4CTLs) are severely reduced in *Myd88-/-* mice. (**A–B**) Representative dot plots of granzyme B (GzB) and T-bet staining on gated CD4+ T splenocytes (left panels) and mean frequencies of T-bet+, IFN-γ+ and GzB+ CD4+ T cells and absolute numbers of GzB+CD4+ T cells (on the right) from (**A**) wild-type (B6) and *Il12p40-/-* mice on day 14 post-infection (pi) and (**B**) wild-type (WT) and *Ifng-/-* mice on day 11 pi. (**C**) Representative dot plots and gating strategy. (**D**) Mean frequencies and (**E**) absolute numbers of GzB+ or NKG2A/C/E+, NKG2D+ and CRTAM+ cells

*Figure 4 continued on next page*

*Figure 4 continued*

among GzB⁺CD4⁺ T cells in the spleen of WT (blue) and *Myd88⁻/⁻* (red) mice at day 13 pi. Bars represent mean values in each group; n=5 individually analyzed mice. Error bars = SEM; ns = non-significant; *p≤0.05; **p≤0.01; ****p≤0.0001 (Student's t-test). Representative dot plots of GzB, NKG2A/C/E, NKG2D and CRTAM staining on CD4⁺ T cells from non-infected animals are shown in *Figure 4—figure supplement 1B*. (**F–G**) Frequencies (left) and absolute numbers (right) of perforin-positive (PRF⁺) cells among CD4⁺ T cells (**F**) and of PRF⁺ cells among GzB⁺CD4⁺ T cells (**G**) in the spleen of non-infected (ctr) or infected (inf.) WT (blue bars) and *Myd88⁻/⁻* (red bars) mice. GzB⁺CD4⁺ T cells were gated as shown in *Figure 1A-D*. Data in (**F**) and (**G**) are compiled from four independent experiments with n=3 animals in each group, each symbol represents an individual analyzed mice. ****p≤0.0001 (Student's t-test). (**H**) Frequency of dead target cells (LPS-induced B cell blasts) in cytotoxic assay, after 14 hr of co-culture with CD4⁺CD44ʰⁱ T cells sorted from WT (blue line) or *Myd88⁻/⁻* (red line) infected mice at day 13 pi. B cells were coated with anti-CD3 (left) or not (right). Mean of triplicates +/- SEM for each E:T ratio point are shown. Data are representative of three independent experiments.

The online version of this article includes the following figure supplement(s) for figure 4:

**Figure supplement 1.** Equivalent frequencies and absolute numbers of total CD4⁺ and CD8⁺ T cells and equivalent frequencies and absolute numbers of CD8⁺ T cells expressing cytotoxic markers in the spleens of infected *Myd88⁻/⁻* and wild-type (WT) mice.

**Figure supplement 2.** Equivalent frequency and absolute number of activated/memory (CD44ʰⁱ) CD4⁺ T cells in wild-type (WT) and *Myd88⁻/⁻* mice infected with *T. cruzi*.

the transcriptional signature of CD4CTLs (*Donnarumma et al., 2016*) was present in sorted CD4⁺ T splenocytes and to what extent the absence of T-cell intrinsic MyD88-dependent signaling would impact the acquirement of this transcriptional program. This signature, previously described in the experimental model of Friend virus infection, distinguishes CD4CTLs from other CD4⁺ T cell subsets, including Th1 and Tfh cells (*Donnarumma et al., 2016*). Importantly, gene set enrichment analysis (GSEA), comparing the transcriptome of sorted WT and *Myd88⁻/⁻* CD4⁺ T cells, revealed that 51 genes of the CD4CTL signature were upregulated in WT CD4⁺ T cells, while 78 were down-regulated in *Myd88⁻/⁻* CD4⁺ T cells (*Figure 5A–C*). We confirmed, by flow cytometry, the upregulation of Blimp-1 (*Prdm1* gene) TF and of the ecto-nucleotidase CD39 (*Entpd1*) in WT CD4⁺GzB⁺ T cells (*Figure 5D and E*). Of note, Blimp-1 is required for the cytotoxic function of CD4CTLs (*Hua et al., 2013*) and CD39 expression level was found to be higher in CD4⁺GzB⁺ T cells than in Tregs (*Figure 5E*).

In order to identify the receptor signaling upstream MyD88 responsible for the establishment of a robust CD4CTL response, we further constructed three groups of mix-BM chimeras, by mixing 1:1 WT and (i) *Il1r1⁻/⁻*, (ii) *Il18ra⁻/⁻* or (iii) *Myd88⁻/⁻* BM cells. A fourth group of mix-BM chimeras, 1:1 WT(B6):WT (B6.SJL) control was also generated. After infection with *T. cruzi*, the percentages and absolute numbers of GzB⁺ cells among CD4⁺ T lymphocytes were analyzed in each chimeric group (gate strategy in *Figure 5—figure supplement 1*). *Figure 5F* shows that the lack of IL-18R signaling in T cells, but not of IL-1R, phenocopied the absence of T cell-intrinsic MyD88 signaling, revealing that IL-18R is a critical pathway upstream MyD88 necessary for the establishment of the CD4⁺GzB⁺ T cell population.

In order to further consolidate the role of IL-18R signaling in the generation of a strong CD4CTL response, we analyzed the expression of PRF as well as of CRTAM, NKG2D and NKG2A/C/E cyto-toxic markers on CD4⁺GzB⁺ T cells from the spleens of *Il18ra⁻/⁻* mice. *Figure 6A* shows that both the frequency and absolute number of PRF⁺CD4⁺ T cells were significantly diminished in the spleens of infected *Il18ra⁻/⁻* compared to WT mice. The same was true for PRF-expressing cells among GzB⁺CD4⁺ T lymphocytes (*Figure 6B*). As observed in infected *Myd88⁻/⁻* mice, the frequency of GzB⁺CD4⁺ T cells and their absolute number were also decreased in infected *Il18ra⁻/⁻* compared to WT mice (*Figure 6C–E*). Absolute numbers of CRTAM⁺, NKG2D⁺ and NKG2A/C/E⁺ GzB⁺CD4⁺ T cells were significantly diminished in *Il18ra⁻/⁻* mice as well, again phenocopying results obtained in *Myd88⁻/⁻* mice (*Figure 6C–E*). The cytotoxic potential of CD4⁺ T splenocytes from *Il18ra⁻/⁻* infected animals was also evaluated by performing redirected cytotoxicity assays with equal numbers of CD4⁺CD44ʰⁱ T cells purified from infected *Il18ra⁻/⁻* or WT mice. Again, as shown for *Myd88⁻/⁻* CD4⁺ T cells, lower levels of target killing were observed when employing purified *Il18ra⁻/⁻* CD4⁺CD44ʰⁱ T cells as effectors (*Figure 6F*). Collectively, the present data demonstrate that T-cell intrinsic IL-18R/MyD88 signaling support the generation and/or survival of CD4⁺ T cells with cytotoxic potential in response to para-site infection, revealing a parallel between the requirements for the development of robust Th1 and CD4CTL responses.

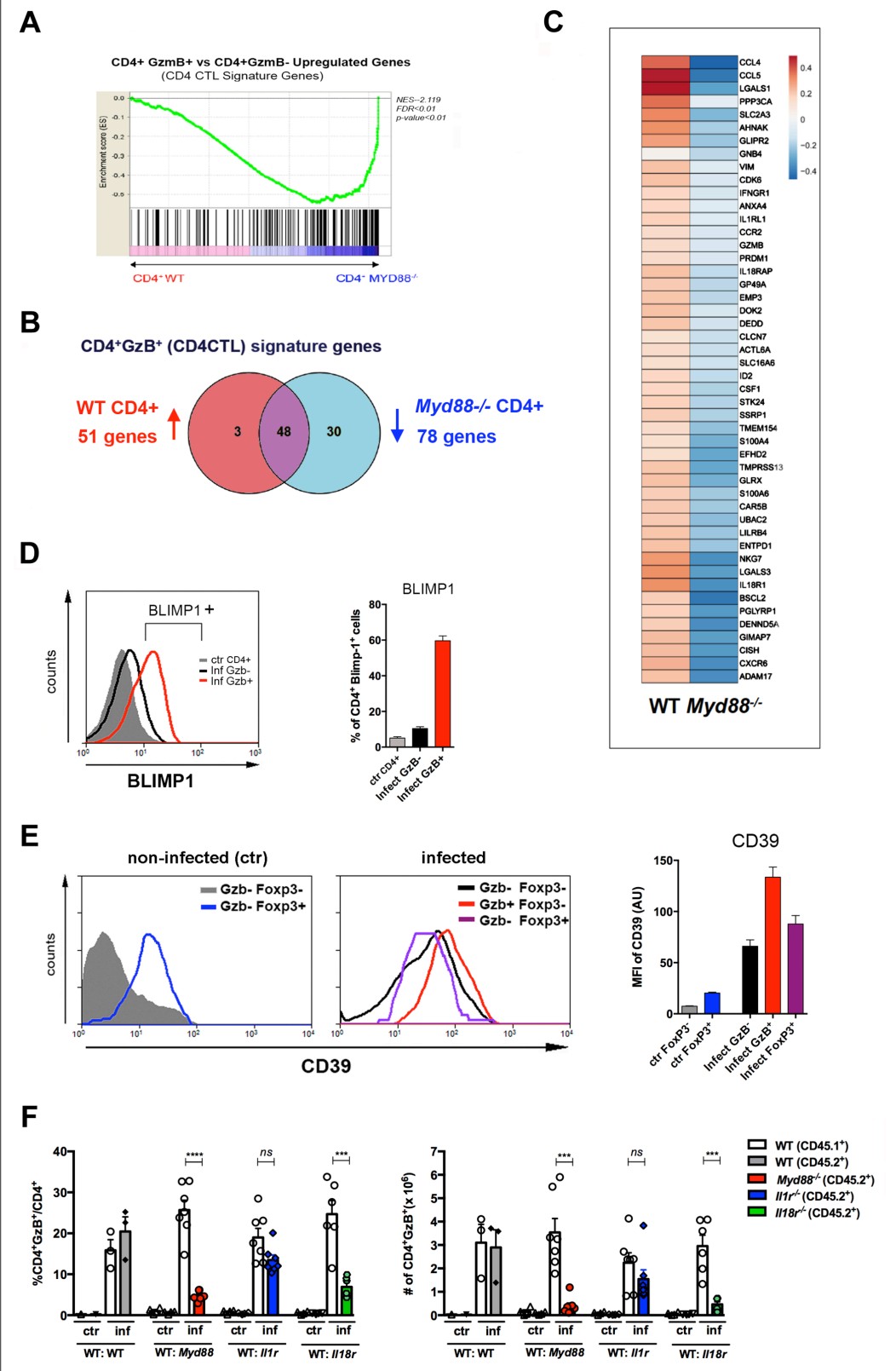

**Figure 5.** The cytotoxic CD4+ T cells (CD4CTL) gene program is absent in MyD88-deficient CD4+ T cells. (**A**) GSEA performed using pre-ranked list of genes expressed in wild-type (WT) or *Myd88−/−* CD4+ T cells sorted from mixed bone marrow (mix-BM) chimeras using the CD4CTL gene signature previously described as gene set (***Donnarumma et al., 2016***). (**B**) Venn diagram of CD4CTL genes upregulated in WT and downregulated in *Myd88-*

*Figure 5 continued on next page*

*Figure 5 continued*

$^{/-}$ CD4$^+$ T cells sorted from chimeric mice. (**C**) Heat map of normalized CD4CTL-signature gene expression in WT or *Myd88*$^{-/-}$ CD4$^+$ T cells sorted from chimeric mice: values indicate mean Z-score, upregulated genes in red and downregulated genes in blue. (**D**) Representative histograms of Blimp-1 expression (right) and mean frequency of Blimp-1$^+$ splenocytes (left) among total gated CD4$^+$ T cells from non-infected mice (ctr, gray shaded area and bar) or among CD4$^+$GzB$^+$ (red line and bar) or CD4$^+$GzB$^-$ (black line and bar) from WT infected mice at day 14 post-infection (pi). Individually analyzed mice, n=4. Error bars = SEM, data are representative of two independent experiments. (**E**) Representative histograms (left and central panel) and mean mean fluorescence intensity (MFI) values of CD39 expression (right panel) on gated FoxP3$^-$ (gray shaded area and bar) or FoxP3$^+$ (blue line and bar) CD4$^+$ T cells from non-infected (ctr) mice (left); GzB$^-$FoxP3$^-$ (black line and bar), GzB$^-$FoxP3$^+$ (purple line and bar) or GzB$^+$FoxP3$^-$ (red line and bar) CD4$^+$ T cells from infected WT mice (center), at day 14 pi; individually analyzed mice (n=4), error bars = SEM; data are representative of four independent experiments. (**F**) Mean frequency (left) and absolute numbers (right) of GzB-expressing cells among CD45.1$^+$ (B6.SJL WT) or CD45.2$^+$ (B6 WT or KO) CD4$^+$ T splenocytes in mix-BM chimeric mice, at day 14 pi; WT:WT control chimera group, n=3; other chimera groups, n=6 or 7. Error bars = SEM, ns = non-significant, ***p≤0.001,****p≤0.0001 (paired Student's t-test). Gating strategy and representative dot plots are shown in *Figure 5—figure supplement 1*. Data are representative of three independent experiments.

The online version of this article includes the following figure supplement(s) for figure 5:

**Figure supplement 1.** GzB expression in CD4$^+$ T cells of mixed bone marrow (mixed-BM) chimeric mice.

## The adoptive transfer of CD4$^+$GzB$^+$ T cells to *Il18ra*$^{-/-}$ mice extended survival to infection

In vivo studies addressing the function of CD4$^+$GzB$^+$ T cells have been hampered by the absence of a stable and abundantly expressed surface-marker of cytotoxicity, which would allow purifying these cells alive. To circumvent this, we employed infected and tamoxifen-treated GzmbCreER$^{T2}$/ ROSA26EYFP reporter mice, from whose spleens CD4$^+$EYFP$^+$ (GzB$^+$) cells were highly purified by flow cytometry and adoptively transferred to infected *Il18ra*$^{-/-}$ mice, following the scheme shown in *Figure 6—figure supplement 1*. The *Il18ra*$^{-/-}$ mouse lineage was previously shown to be very susceptible to infection with *T. cruzi* (*Oliveira et al., 2017*), and display lower numbers of CD4$^+$GzB$^+$ and CD4$^+$PRF$^+$ T cells with cytotoxic potential, as demonstrated above. The transfer of highly purified CD4$^+$GzB$^+$(EYFP$^+$) T cells to *Il18ra*$^{-/-}$ mice significantly extended their survival rate without, however, decreasing the parasite load in the hearts (*Figure 6G*). In a previous work, we have demonstrated that the transfer of total WT CD4$^+$ T cells protects infected *Il18ra*$^{-/-}$ mice, and this property was attributed to Th1 cells, which are also diminished in this mouse lineage (*Oliveira et al., 2017*). Since infected *Ifng*$^{-/-}$ mice have the same frequency of CD4$^+$GzB$^+$ T cells as WT mice in their spleens (*Figure 4B*), we tested whether adoptive transfer of CD4$^+$ T cells sorted from *Ifng*$^{-/-}$ mice would also be able to extend the survival of recipient *Il18ra*$^{-/-}$ mice. As shown in *Figure 6—figure supplement 2*, this was indeed the case. Although IFN-γ expression by CD4$^+$ T cells is necessary for decreasing parasitemia (*Oliveira et al., 2017*), the present result indicated that the secretion of IFN-γ by adoptively transferred CD4$^+$ T cells is not absolutely required for extending the survival time of infected *Il18ra*$^{-/-}$ recipients. Together, these data demonstrate the beneficial function of the CD4$^+$GzB$^+$ T cell subset against mortality caused by an intracellular parasite, which is independent of decreasing heart parasitism.

## The great majority of CD4$^+$ T cells in the infected myocardium express the CD4CTL phenotype and increased levels of immunoregulatory markers

Previous studies in viral infection models suggested that CD4CTLs are present in tissues where pathogen replication occurs (*Marshall et al., 2017*; *Takeuchi et al., 2016*). Since *T. cruzi* heavily infects the heart, we performed intravascular staining with anti-CD4 mAb and demonstrated that tissue-infiltrating CD4$^+$ T cells are abundant in the myocardium of infected mice (*Figure 7—figure supplement 1A, B*). As shown in *Figure 7A and B* and *Figure 7—figure supplement 2*, CD4$^+$GzB$^+$ T cells infiltrating the cardiac tissue constitute the vast majority of CD4$^+$CD44$^{hi}$ T cells, indicating the enrichment of this subset among activated CD4$^+$ T cells in the heart compared to the spleen. At the peak time-point of parasite load in the heart, CD4$^+$GzB$^+$ T cells in the myocardium express the LAMP-1 degranulation marker at levels comparable to those of CD8$^+$ T cells (*Figure 7C and D* and

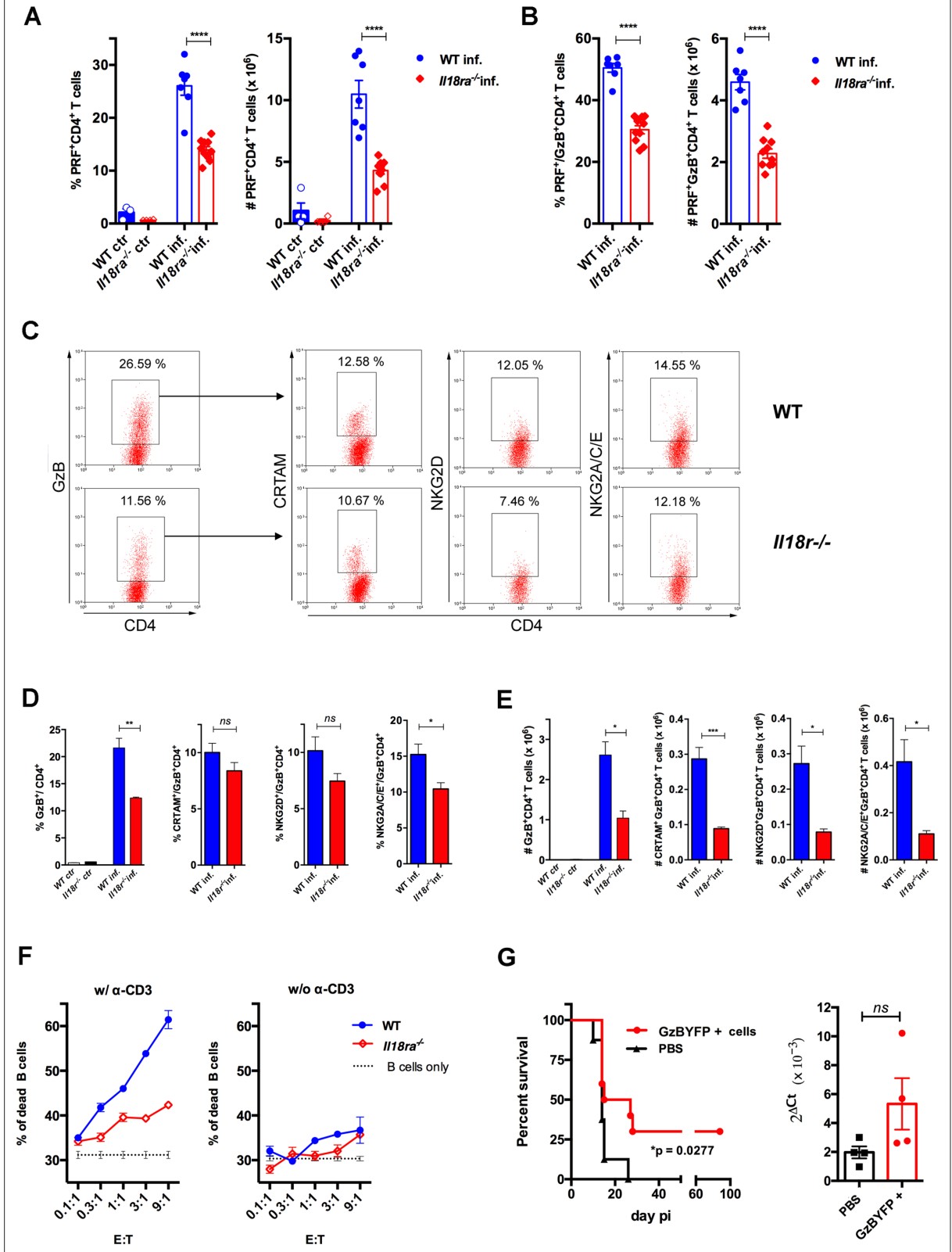

**Figure 6.** Cytotoxic CD4+ T cells (CD4CTLs) are severely reduced in *Il18ra-/-* mice. (**A and B**) Frequencies and absolute numbers of perforin-positive (PRF+) cells among CD4+ T cells (**A**) and of PRF+ cells among GzB+CD4+ T cells (**B**) in wild-type (WT) (blue bars) and *Il18ra-/-* (red bars) spleens of non-infected (ctr) or infected (inf.) mice. Data are compiled from two independent experiments with n=4–5 animals in each group, each symbol represents an individual mouse. (**C**) Representative dot plots of GzB, CRTAM, NKG2D and NKG2A/C/E staining in WT and *Il18ra-/-* mice. (**D**) Mean frequencies and

*Figure 6 continued on next page*

Figure 6 continued

(**E**) absolute numbers of GzB⁺ or CRTAM⁺, NKG2D⁺ and NKG2A/C/E⁺ cells among CD4⁺ and CD4⁺GzB⁺ T cells, respectively. (**A–E**) CD4⁺ T cells gated as in **Figure 1A–D**; n=4 individually analyzed mice in each group; error bars = SEM, *p≤0.05, **p≤0.01, ***p≤0.001, ns = non-significant (Student's t-test). Data are representative of two independent experiments. (**F**) Frequency of dead target cells (LPS-induced B cell blasts) in cytotoxic assay, after 14 hr of co-culture with CD4⁺CD44ʰⁱ T cells sorted from WT (blue line) or *Il18ra⁻/⁻* (red line) infected mice at day 13 post-infection (pi). B cells were coated with anti-CD3 (left) or not (right). Mean of triplicates +/- SEM for each E:T ratio point are shown. (**G**) Survival curve (left) and parasite load in the heart (right) of infected *Il18ra⁻/⁻* mice, adoptively transferred with CD4⁺GzBYFP⁺ cells at day 7 pi (red curve and bar), or not (black curve and bar); time-line in **Figure 6—figure supplement 1**; n=10 mice in each group; p=0.0277 Log-rank (Mantel-Cox) test; survival data are compiled from two independent experiments; ns = non-significant (Student's t-test).

The online version of this article includes the following figure supplement(s) for figure 6:

**Figure supplement 1.** Time-line of the adoptive transfer experiment shown on **Figure 6G**.

**Figure supplement 2.** The adoptive transfer of *Ifng⁻/⁻* CD4⁺ T cells increased survival to infection.

**Figure 7—figure supplement 1C**). Importantly, the great majority of heart-infiltrating CD4⁺GzB⁺ T cells expresses PRF, as well as the CRTAM and NKG2D cytotoxic markers (**Figure 7E and F**). On the other hand, only few CD4⁺GzB⁺ T cells express NKG2A/C/E in the myocardium (**Figure 7—figure supplement 1D**, left). These data indicate a selective migration to, or retention in, the infected heart of double positive CRTAM⁺NKG2D⁺ CD4⁺GzB⁺ T cells, although both NKG2A/C/E⁺ and NKG2D⁺ cells are present at a similar ratio among CD4⁺GzB⁺ T cells in the spleen. (**Figure 3—figure supplement 3**). On the other hand, high percentages of NKG2A/C/E⁺CD8⁺GzB⁺ T cells were found both in the heart (**Figure 7—figure supplement 1D**, right) and in the spleen (**Figure 3—figure supplement 4A**) of infected mice.

We next investigated the ratio of cells expressing the immunoregulatory markers Lag-3, Tim-3, CD39 and PD-1 among different CD4⁺ T cell subsets of infected mice: naïve cells (CD44ˡᵒGzB⁻PRF⁻), activated non-cytotoxic (CD44ʰⁱGzB⁻PRF⁻) or activated cytotoxic (CD44ʰⁱGzB⁺PRF⁺) cells. The intensity of expression of these markers in each cell subpopulation was also analyzed. Results in **Figure 7—figure supplement 3**, show that CD44ʰⁱGzB⁺PRF⁺ CD4⁺ T cells compose the subset of activated cells among which the higher frequencies of cells expressing these markers are found, both in the spleen and in the heart, at day 14 pi. The only exception was the equal ratio of cells expressing PD-1, at equivalent levels, when comparing splenic CD44ʰⁱGzB⁻PRF⁻ and CD44ʰⁱGzB⁺PRF⁺ CD4⁺ T cells. Similarly, equivalent percentages of cells expressing PD-1 among CD44ʰⁱGzB⁻PRF⁻ and CD44ʰⁱGzB⁺PRF⁺ CD4⁺ T cells were found in the heart. However, the intensity of expression of the PD-1 marker was significantly higher among CD44ʰⁱGzB⁺PRF⁺ (CD4CTLs) compared to CD44ʰⁱGzB⁻PRF⁻ CD4⁺ T cells infiltrating the cardiac tissue. Furthermore, we analyzed whether these parameters differ between CD4CTLs in the spleen and in the heart (**Figure 7—figure supplement 4**). Of note, while mean fluorescence intensity (MFI) values of Tim-3, CD39 and PD-1 expression were increased on CD4CTLs in the heart compared to CD4CTLs in the spleen, Lag-3 expression levels were decreased on CD4CTLs infiltrating the myocardium. Furthermore, only the expression of PD-1, but not of Lag-3, Tim-3 or CD39, is upregulated on CD8CTLs in the heart compared to the spleen (**Figure 7—figure supplement 4**), an indication that the *T. cruzi*-infected cardiac tissue is a less so-called exhaustion-inducing environment compared to certain tumors.

## The frequency of CD4⁺PRF⁺ T cells correlates with cardiomyopathy in Chagas patients

Finally, we investigated the presence of PRF⁺CD4⁺ T cells in the circulation of patients infected with *T. cruzi* and suffering from chronic Chagas cardiomyopathy (CC). In order to ascertain the specificity of CD4⁺ T cells against *T. cruzi*, PBMC from CC patients or from healthy donors (HD) were incubated (or not) with trypomastigote-derived antigens (trypo) and analyzed for the expression of perforin by flow cytometry (**Figure 7—figure supplement 5**). As shown in **Figure 7G and H**, after incubation with trypo, both the expression level of PRF (MFI) and the percentage of PRF⁺CD4⁺ T cells increased among CD4⁺ T cells from CC patients, but not from HD controls. Of note, CD4⁺ T cells expressing PRF were found at higher frequency in CC patients than in HD (**Figure 7I**). We next performed correlative analyses between the frequency of PRF-expressing CD4⁺ T cells in PBMC from CC patients and two echocardiographic parameters, usually employed to evaluate cardiac function in CC patients: left ventricular ejection fraction (LVEF) and left atrium diameter. As shown in **Figure 7J and a** clear

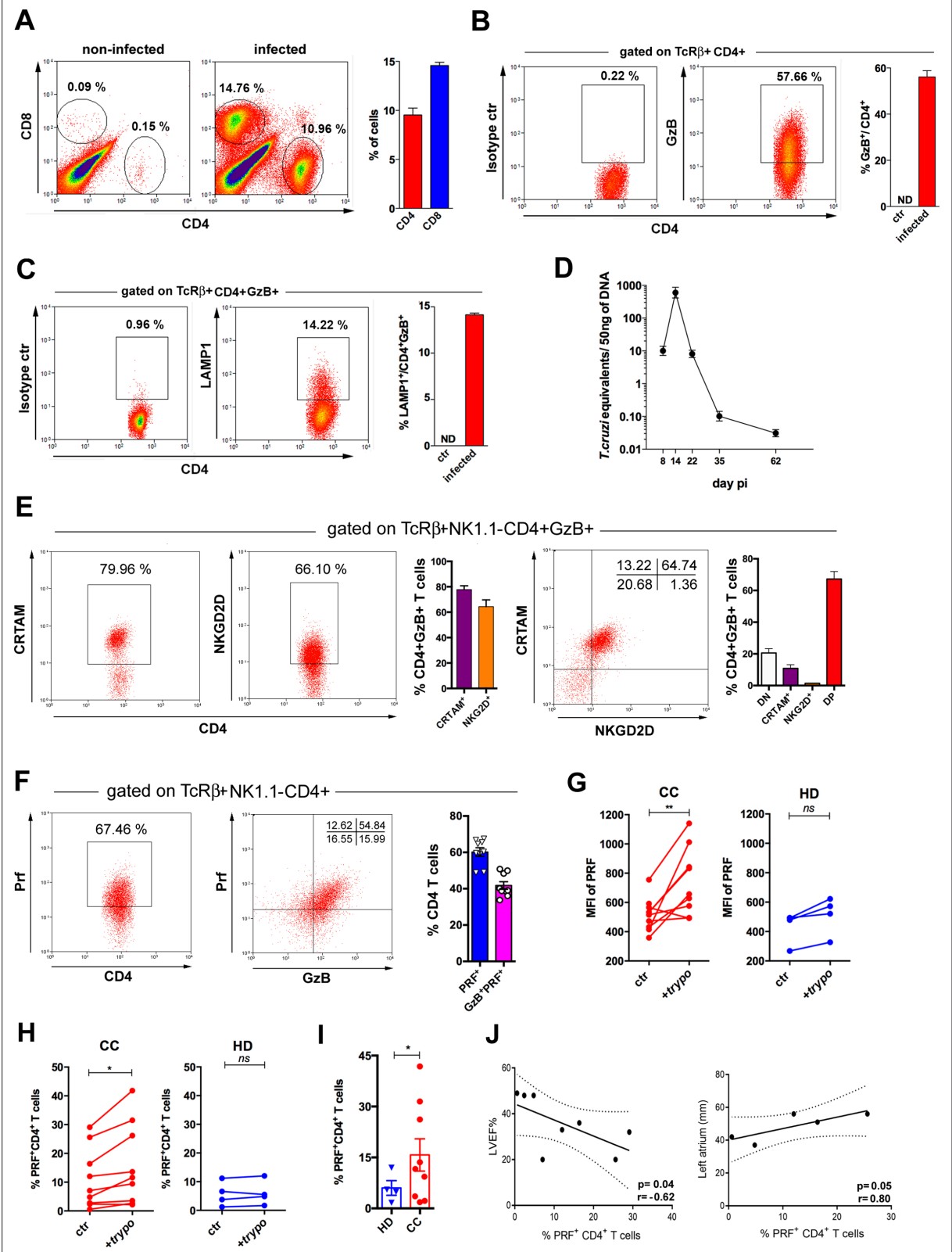

**Figure 7.** Cells with cytotoxic CD4+ T cell phenotype predominate in the cardiac tissue of infected mice and are at higher frequencies in patients with Chagas cardiomyopathy (CC). (**A**) Representative dot plots (gated as in *Figure 7—figure supplement 1A*), and mean frequency of CD8+ and CD4+ T cells infiltrating the heart of B6 infected mice at day 14 post-infection (pi). (**B and C**) Representative dot plot and mean frequency of (**B**) granzyme B-positive (GzB+) and (**C**) LAMP-1+ (CD107a+) cells, among gated CD4+ T cells infiltrating the infected heart; ctr = non-infected, ND = non-detected. (**D**)

*Figure 7 continued on next page*

*Figure 7 continued*

Mean parasite load in the cardiac tissue at different time points pi, obtained by real-time PCR, of individually analyzed mice as described in Methods; data are representative of at least eight (**A and B**) or two (**C and D**) independent experiments. (**E**) Representative dot plots and mean frequency of CRTAM+ and NKG2D+ cells among gated intracardiac CD4+GzB+ T cells. DN and DP=CRTAM and NKG2D double negative and doulbe positive cells, respectively. Individually analyzed mice, n=4; error bars = SEM; data are representative of four independent experiments. (**F**) Representative dot plots and mean frequency of perforin-positive (PRF+) and GzB+PRF+ cells among intracardiac CD4+ T cells; data are compiled from two independent experiments with n=4 or 5 animals in each group, each symbol represents an individual analyzed mice. (**G**) Intensity of PRF staining (mean fluorescence intensity [MFI]) and (**H**) Frequency of PRF+ cells among CD4+ T cells in PBMCs from patients with chronic Chagas disease cardiomyopathy (CC; n=10) and healthy donors (HD; n=4) cultured overnight in medium alone (ctr) or in the presence of trypomastigote antigens (+trypo); *p≤0.05, **p≤0.01, ns = non-significant; data analyzed by paired Student's t test. (**I**) Frequency of PRF+ cells among CD4+ T cells in the peripheral blood of patients with chronic Chagas disease cardiomyopathy (CC; n=10) and healthy donors (HD; n=4) cultured ON in the presence of trypomastigote antigens. Error bars = SEM; *p≤0.05, Student's t test. Gating strategy and representative dot plots are shown in *Figure 7—figure supplement 5*. (**J**) Correlation analysis between the percentage of CD4+PRF+ T cells among CD4+ T cells in PBMCs from CC patients and left ventricular ejection fraction (LVEF) (left panel) or left atrium diameter (right panel); indicated p and r values were calculated using Pearson correlation test.

The online version of this article includes the following figure supplement(s) for figure 7:

**Figure supplement 1.** CD4+ and CD8+ T cells in the heart of mice infected with *T. cruzi*.

**Figure supplement 2.** GzB+CD4+ T cells are enriched among activated/memory CD4+ T cells in the heart compared to the spleen of mice infected with *T. cruzi*.

**Figure supplement 3.** Cytotoxic CD4 +T cells express the higher levels of immunoregulatory molecules.

**Figure supplement 4.** The expression of PD-1, CD39 and Tim-3, but not of Lag-3, is increased on cytotoxic CD4+ T cells (CD4CTLs) infiltrating the heart, compared to their levels in the spleen.

**Figure supplement 5.** CD4+ T cells expressing perforin (PRF) are found in the circulation of patients suffering from chronic Chagas cardiomyopathy (CC).

**Figure supplement 6.** Predicted model for the generation of cytotoxic CD4+ T cells (CD4CTLs) in response to *T. cruzi* infection.

inverse correlation between the frequency of CD4+PRF+ T cells and LVEF, as well as a direct correlation between the frequency of these cells and the left atrium diameter was found. These correlations suggest that CD4CTLs might play a pathogenic role in the chronic phase of Chagas disease, although the alternative possibility remains that the higher levels of CD4CTLs reflect elevated parasite burden and/or inflammation in the heart, without a direct involvement of this cell subset in the pathology. Together, data obtained here in CC patients and in the mouse experimental model point to the heart as an important target-organ for CD4+ T cells with cytolytic potential generated in response to infection with *T. cruzi*.

## Discussion

Here we show for the first time that CD4CTLs are abundantly present in the spleen of mice infected with *T. cruzi* and that such phenotype predominates among CD4+ T cells infiltrating the infected cardiac tissue. The expression of LAMP-1 degranulation marker by CD4+GzB+ T cells in the heart and in the spleen indicates that these cells may play important effector functions in both organs. GzB+PRF+CD4+ T cells present in the *T. cruzi*-infected hearts express CRTAM and NKG2D cytotoxicity markers, but differently from CD4CTLs found in the lungs of mice infected with IAV, do not express NKG2A/C/E molecules (*Marshall et al., 2017*). This interesting dissimilarity requires further investigation and could be due to differential regulation exerted by tissue-specific factors or, alternatively, by different innate responses influencing CD4CTL differentiation, which would be distinctly triggered by viral and parasite pathogens.

CD4+ T cells purified from the spleen of infected mice killed APCs presenting parasite-derived antigens in vitro, in a GzB- and PRF-dependent process. In vivo Ag-specific cytotoxicity against parasite-derived I-A^b-restricted epitopes was also observed. It is expected that killing of infected APCs in the spleen should lead to a decrease in the activation of effector T cells, dampening the exuberant immune response against the parasite. Therefore, the cytotoxic action of CD4CTLs may contribute to immunoregulation, avoiding the development of exacerbated immunopathology that can cause mortality in the acute phase of the infection with *T. cruzi*. Favoring this hypothesis, we showed that adoptive transfer of CD4+GzB+ T cells to susceptible *Il18ra^-/-* mice, which have lower levels of these cells, significantly increased their survival to infection without, however, affecting parasite loads. The

incapacity of transferred CD4CTLs in decreasing parasite loads in the heart might be due to two facts: (i) cytotoxicity mediated by CD4CTLs is MHC class II-restricted, but only heart endothelial cells express MHC class II molecules, while the infected cardiac muscle cells remain MHC II[-] (*Reis et al., 1993*); (ii) as shown here, only a small fraction of CD4[+] T cells with cytolytic phenotype produce IFN-γ, the main cytokine inducing trypanosomicidal function in macrophages and cardiomyocytes (*Machado et al., 2000*). Accordingly, adoptive transfer of highly purified *Ifng*[-/-] CD4[+] T cells (among which the same frequency of GzB[+] cells were found) also resulted in the significantly increase of the survival time of infected *Il18ra*[-/-] mice, but did not decrease parasite levels (*Oliveira et al., 2017*). The data discussed above, along with the fact that the parasite load is not decreased in the recipient mice indicate that IFN-γ is not involved in the extended survival granted by adoptive transfer of CD4[+]GzB[+] T cells and exclude a role for Th1 cells, which are known to exert an important microbicidal function through the production of IFN-γ (*Oliveira et al., 2017*). Together with our previous work, the current results point to complementary roles exerted by CD4CTL and Th1 cells. Both cell types might be required for survival against infection with *T. cruzi*, working together to establish a balance between the Th1-dependent control of parasite load on one side, and CD4CTL-mediated dampening of the immunopathology resulting from this process, on the other. In fact, it is known that IFN-γ plays a central but dual role in Chagas disease, having both important microbicidal and pathogenic functions (*Ferreira et al., 2014*). The equilibrium between anti-parasitic and anti-inflammatory responses underlies the establishment of the asymptomatic form of chronic infection (*Nunes et al., 2018*), in which increased levels of CD4[+]GzB[+] T cells have been found in humans (*Keesen et al., 2012*). Therefore, our results are in accordance with an immunoregulatory role played by CD4CTLs, likely through the GzB/PRF/FasL-mediated killing of infected APCs in an IFN-γ-independent manner, although it is not possible to formally attribute the beneficial role of adoptively transferred CD4CTLs exclusively to their cytolytic action.

Of note, we also show here that most CD4[+]GzB[+]PRF[+] T cells express surface co-inhibitory receptors, such as Lag-3 and Tim-3, as well as the ecto-ATPase CD39, at particularly high levels. These regulatory molecules, typically expressed by Treg, Tr1 cells and terminally differentiated or exhausted CD8[+] or CD4[+] T cells, have been implicated in the immunoregulatory functions of these cell subsets (*Anderson et al., 2016*; *Mascanfroni et al., 2015*; *Park et al., 2019*). Although almost the totality of CD4[+]GzB[+] T cells express T-bet, only a minority of these cells produce IFN-γ. Most CD4[+]GzB[+] T cells also express Eomes and/or Blimp-1, two transcription factors, which are known to be involved in T cell exhaustion. Whether the repression of IFN-γ in CD4[+]GzB[+] T cells is mediated by Eomes and/or Blimp-1, as evidentiated in CD4[+] T cells during infection with *Toxoplasma gondii* (*Hwang et al., 2016*), remains to be determined. Thus, the phenotype of a large fraction of CD4[+]GzB[+]PRF[+] T cells developing in response to infection with *T. cruzi* is typical of terminal differentiated/exhausted cells. Nevertheless, it is important to note that highly functional effector T cells can also express inhibitory receptors and despite exhaustion being often seen as a dysfunctional state, it has been shown that the expression of inhibitor molecules allows T cells to persist and to contribute to the contention of chronic viral infections without causing immunopathology (reviewed in *Wherry, 2011*).

Chronic CC usually develops in around 30% of infected individuals after decades, probably reflecting an imbalance between parasite and host immune response (*Nunes et al., 2018*). Although the exact mechanism remains to be elucidated, the association of NK, CD8CTL and GzA-expressing CD4[+] T cells with the development of chronic carditis has been indicated (*Ferreira et al., 2017*; *Menezes et al., 2012*; *Nunes et al., 2018*). Since CD4CTLs recognize the cognate antigen in the context of MHC class II molecules, the lysis of infected MHC II[+] endothelial cells in the heart might trigger immunopathology in the chronic phase, in contrast to the possible immunoregulatory role due to the killing of infected APCs in the spleen, during the acute phase of infection, as discussed above. In the present study, we found that the frequency of parasite-specific CD4[+]PRF[+] T cells is increased in the circulation of chronic cardiac patients with Chagas disease (CC) when compared to HD. A correlation between the frequency of PRF[+]CD4[+] T cells and two echocardiographic parameters employed for evaluating myocarditis in CC patients was also encountered. It is tempting to speculate that the loss of regulatory function exerted by different CD4[+] T cell subsets, including CD4CTLs, might lead to the chronic pathology. Alternatively, the increased levels of CD4CTLs in chronic patients might be reflecting parasite burden and/or tissue inflammation without exerting a pathogenic function. Therefore, further studies are necessary for fully defining the role of tissue-infiltrating CD4CTLs in Chagas

disease. Interestingly, NKG2D, expressed by most CD4CTLs in the heart, can mediate both activating and inhibitory signals (*Wensveen et al., 2018*). Importantly, a polymorphism in one of the NKG2D ligands, the major histocompatibility complex class I–related chain A gene (MICA) molecule, was associated with left ventricular systolic dysfunction in patients with chronic Chagas myocarditis (*Ayo et al., 2015*).

We have shown here that CD4+GzB+ T cells originating in response to *T. cruzi* infection largely outnumber IFN-γ-producing CD4+ T cells and that their appearance in the spleen follows the same kinetics of classic Th1 cells, indicating thereby that differentiation of CD4CTLs is not a late event during infection. Of note, more than 60% of CD4+GzB+ T cells have downregulated CD27 expression in the spleen at day 14 pi, a characteristic of terminally differentiated T cells (*Takeuchi and Saito, 2017*). We have also shown here that the majority of CD4+GzB+ T cells express T-bet, Eomes and Blimp-1. Although Eomes seems not to be essential for CD4CTLs in certain studies (*Donnarumma et al., 2016*; *Kotov et al., 2018*), it was proposed to be the master regulator for CD4CTLs and Tr1 cells by others (*Curran et al., 2013*; *Eshima et al., 2012*; *Qui et al., 2011*; *Zhang et al., 2017*). The CD4+GzB+ T cell population described herein presents some characteristics in common with Tr1 cells. However, we only detected a very low frequency of CD4+GzB+ T cells co-expressing Lag-3 and CD49b (*Gagliani et al., 2013*) or IL-10. In this regard, data from *T. cruzi* infection differs from those observed during infection with other intracellular parasites such as *T. gondii* and certain *L. major* strains, in which high frequencies of CD4+ T cells producing both IL-10 and IFN-γ were described (*Anderson et al., 2007*; *Jankovic et al., 2007*). In contrast with ThPOK- CD4CTLs that accumulate in the gut (*Cheroutre and Husain, 2013*; *Mucida et al., 2013*; *Reis et al., 2013*), CD4+GzB+ T cells in the *T. cruzi*-infected spleen express ThPOK at the peak of infection. Interestingly, other studies observed the absence of ThPOK downregulation in mouse and human CD4CTLs (*Mittal et al., 2018*; *Serroukh et al., 2018*). On the other hand, we also detected *Runx3d* expression in GzB+ CD4+ T cells, although it was not possible to compare it to the levels found in the Th1 cell subset (*Djuretic et al., 2007*). Thus, different subpopulations of CD4CTL possibly exist, in which the interplay between ThPOK and RUNX3 expression might be differentially regulated.

Although many authors consider CD4CTLs merely functional variants of Th1 cells, in some models CD4CTLs fail to fit Th1-defining criteria, such as the dependence on IL-12, STAT4 or T-bet for differentiation (*Brown et al., 2009*; *Curran et al., 2013*; *Hua et al., 2013*; *Kotov et al., 2018*). Nonetheless, in agreement with other studies (*Cooper et al., 2004*; *Krueger et al., 2021*; *Tagawa et al., 2016*; *Xie et al., 2010*), we showed here that CD4+GzB+ T cells are scarcely generated in the absence of IL-12p40. Importantly, we demonstrated that the expansion/survival of CD4CTLs also depends on T-cell intrinsic IL-18R/MyD88 signaling, which we have previously shown to induce cell cycling, anti-apoptotic and Th1-signature transcripts (*Oliveira et al., 2017*). Notably, several genes of the CD4CTL signature, which distinguishes this subset from Th1 and Tfh cells (*Donnarumma et al., 2016*), were expressed by WT, but not by *Myd88-/-* CD4+ T cells from infected mix-BM chimeras. Therefore, analogous to its action on Th1 cells, we showed here that T-cell intrinsic IL-18R/MyD88 signaling contributes for the reinforcement and/or stabilization of CD4+ T cell commitment into the CD4CTL phenotype. IFN-γ is another key cytokine involved in Th1 differentiation and *Ifng-/-* mice are extremely susceptible to infection with *T. cruzi* (*Campos et al., 2004*). Interestingly, we found no difference in the frequency of CD4+GzB+ T cells between WT and *Ifng-/-* infected mice, in accordance with a study showing that IFN-γ had no impact on the ability of CD4 effectors to acquire cytolytic activity (*Brown et al., 2012*). However, both GzB+ and total CD4+ T-cell numbers were significantly lower in the infected *Ifng-/-* animals, indicating that IFN-γ might also play a role in the expansion/survival of both Th1 and CD4CTLs in our system. It is known that IFN-γ and IL-18R participate in a reciprocal positive feedback loop in Th1 differentiation process: while IFN-γ is required for IL-12-driven upregulation of IL-18R expression (*Nakahira et al., 2001*; *Smeltz et al., 2001*), IL-18 is a major IFN-γ-inducing factor. Therefore, the absence of IL-18R upregulation in CD4+ T cells of *Ifng-/-* mice would prevent the expansion of both CD4+ T cell subpopulations. Thus, in *T. cruzi* infection model, IL-12, IFN-γ and IL-18R/MyD88 signaling act as key pathways at early check-points in the generation of CD4+GzB+ T cells with cytotoxic potential, suggesting that these cells might share common early differentiating steps with the Th1 subset. It is not excluded, however, that different precursors of CD4CTL and Th1 cells might exist (*Takeuchi and Saito, 2017*). A cartoon summarizing our results and cited works published by others is shown in *Figure 7—figure supplement 6*. Of note, in humans, the higher production of IL-18

due to gene polymorphisms has been correlated to protection against chronic CC (*Leon Rodriguez et al., 2016*; *Nogueira et al., 2015*).

It is possible that differentiation into CD4CTLs could also be influenced by pathogen-derived factors. Very recently, Krueger et al have shown that a subset of IL-12-dependent Th1-derived cytotoxic cells was present in the spleen of *Il18⁻/⁻* mice at later stages of infection with *Salmonella enterica* (*Krueger et al., 2021*). Whether that result was due to compensatory mechanisms, such as the known exacerbated IL-12 production by infected *Il18⁻/⁻* mice (*Monteforte et al., 2000*), or to distinct innate responses, differentially induced by phagosomal bacteria and cytosolic parasites, remains to be determined. Of note, other discrepancies between results obtained with *Il18⁻/⁻* versus *Il18ra⁻/⁻* mouse lineages have been described in different models in the past (*Gutcher et al., 2006*; *Lewis and Dinarello, 2006*). It is known that different classes of pathogens can trigger diverse APC responses and, therefore, elicit pathogen-tailored Th programs (*Kara et al., 2014*; *Kiner et al., 2021*). Previous works have reported the immunoregulatory action of *T. cruzi*-derived PAMPs on CD4⁺ T cell activation, either by directly acting as co-stimulatory signals for T cells or by modulating APC functions (*Bellio et al., 1999*; *DosReis et al., 2002*; *Medeiros et al., 2007*). It would be interesting to test *T. cruzi*-derived molecules on their capacity to favor the generation of CD4CTLs.

In summary, the present work is the first to describe a CD4CTL subset abundantly present in the *T. cruzi*-infected cardiac tissue, bringing novel pieces to the scenario of the immune response against a major human intracellular parasite. Importantly, our data disclose a mechanistic framework involved in the generation of a robust CD4CTL response, whose control and optimization may provide new therapeutic and vaccination strategies to combat infection with intracellular pathogens and chronic diseases resulting thereof.

# Materials and methods
## Mice
C57BL/6 (RID:IMSR_JAX:000664), B6.SJL (RRID:IMSR_JAX:002014), and F1 (B6 × B6.SJL) mice were from the Universidade Federal Fluminense. *Myd88⁻/⁻* (RRID:MGI:5447806) mice were generated by Dr. S. Akira (Osaka University, Japan); *Il12p40⁻/⁻* (RRID:IMSR_JAX:002693) and *Ifng⁻/⁻* (RRID:IMSR_JAX:002287) mice were purchased from USP; *Il18ra⁻/⁻* (RRID:IMSR_JAX:004131) and *Il1r1⁻/⁻* (RRID:IMSR_JAX:003245) mice from JAX Mice, USA. GzB reporter mice, GzmbCreER$^{T2}$/ROSA26EYFP, previously described (*Bannard et al., 2009*) were provided by Dr Ricardo Gazinelli (UFMG, BH, Brazil). Mice were on the C57BL/6 background, except GzmbCreER$^{T2}$/ROSA26EYFP, which were on mixed Sv129/B6 background and backcrossed to B6 for 5 generations. In each experiment, only male mice of the same age (6- to 8-week-old) were used. Mice were housed at a maximum of 5 animals per cage, in forced-air-ventilated micro-isolation rack system, in a room at 21–23°C under a 12/12 hr light/dark cycle and provided with sterilized water and chow ad libitum. All mouse experiments were conducted in accordance with guidelines of the Animal Care and Use Committee of the Federal University of Rio de Janeiro (Comitê de Ética do Centro de Ciências da Saúde CEUA - CCS/UFRJ, license: IMPPG022).

## Experimental infection
Bloodstream trypomastigotes of the Y strain of *T. cruzi* (2 × 10³ cells/0.2 mL) were inoculated intraperitoneally (ip). Y-OVA strain trypomastigotes (*Gomes-Neto et al., 2018*) were obtained from LLC-MK2 infected cultures and 2 × 10⁶ trypomastigotes (in 0.2 mL phosphate-buffered saline; PBS) were inoculated ip. Rare outliers (mice in which the infection was not properly established) were identified by the lower size of the spleen (total cell number) and lower percentage of GzB⁺CD8⁺ T cells (inferior of 50%) in the spleen. These outlier mice were excluded from all analyses. Mice found at the moribund condition (presenting unambiguous signals that the experimental endpoint has been reached), were euthanized.

## Generation of BM chimeras
Acidic drinking water (pH 2.5–3.0) was given for 8 days to recipient mice prior to irradiation with a lethal single dose of 800 rad (TH780C - Theratronics, Canada). The day after irradiation, BM cells were transferred and mice were treated with neomycin sulfate (2 mg/mL) in the drinking water for 15 days. Marrow was harvested from the femur and tibia of B6.SJL (CD45.1⁺) and B6, *Myd88⁻/⁻, Il1r1⁻/⁻* or

*Il18ra*[−/−] (CD45.2[+]) mice. For mixed BM chimeras, CD45.1[+] and CD45.2[+] BM cells were transferred in a 1:1 ratio (2 × 10[6] cells each) to irradiated mice. In all experiments, mice were infected 6–8 weeks after cell transfer, when reconstitution was achieved.

## Human population

A total of thirteen volunteers were enrolled in this study. Nine volunteer patients had positive specific serology for *T. cruzi* and displayed the chronic CC with right and/or left ventricular dilation, global left ventricular dysfunction, alterations in the cardiac electric impulse generation and conduction upon electrocardiogram, chest x-rays and echocardiography; four individuals displayed negative specific serological tests for Chagas disease and were included as control group (HD). Patients were from Chagas disease endemic regions within Minas Gerais, Brazil, and were evaluated at the outpatient clinic of the Universidade Federal de Minas Gerais. HD were from Belo Horizonte, which is a non-endemic region for Chagas disease. We excluded from our study individuals with any other chronic inflammatory diseases, diabetes, heart/circulatory illnesses or bacterial infections. All individuals included in this work were volunteers and treatment and clinical care were offered to all patients, as needed, despite their enrollment in this research project. All participants provided written informed consent for participation in the study. This cross-sectional study is part of an extended project evaluating biomarkers of cardiomyopathy development in Chagas disease, which has the approval of the National Committee of Ethics in Research (CONEP#195/2007) and are in accordance with the Declaration for Helsinki.

## PBMC flow cytometry

Peripheral blood samples were collected in heparin tubes. PBMCs were obtained by separating whole blood over Ficoll (Sigma-Aldrich, St. Louis, MO), as previously described (*Menezes et al., 2012*). Cells were washed and resuspended in RPMI 1640 medium (Gibco, Rockville, MD) supplemented with 5% heat-inactivated AB human serum, antibiotics (penicillin, 200 U/mL; and streptomycin, 0.1 mg/mL) and L-glutamine (1 mM) (Sigma-Aldrich) at a concentration of 1 × 10[7] cells/mL. Cells from each volunteer were cultured on 96-well plates (Costar, Corning Incorporated, Corning, NY) in 200 μL of medium alone (ctr) or containing 20 μg/mL of trypomastigote antigen (trypo) for 18 hr. Trypo was obtained from trypomastigotes purified from VERO cell cultures: parasites were washed, freeze-thawed for 3 cycles, followed by 5 cycles of 5 min each of sonication using 30 s between them. Samples were ultracentrifuged at 5000 g for 60 min, at 4°C. Supernatants were collected, dialyzed for 24 hr and protein quantification performed by Bradford method. Cells (2 × 10[5]) were incubated with PerCP-anti-CD4 (clone A161A1) antibody (Biolegend) for 15 min at 4°C, washed in PBS containing 1% bovine serum albumin, and fixed by a 20 min incubation with a 2% formaldehyde. After washing with PBS, cells were permeabilized with 0.5% saponin for 15 min and intracellularly stained with PE-anti-perforin (clone B-D48) antibody (Biolegend) for 20 min, washed twice with 0.5% saponin solution, resuspended in PBS and analyzed. A minimum of 100,000 gated events from each sample were collected and analyzed using FlowJo software.

## Flow cytometry of mouse cells

Reagents and antibodies: PB-anti-CD45.1, APC-Cy7-anti-CD45.2, FITC- PE- or BV-605-anti-CD4, PE-anti-CD8, PE-Cy7-anti-NK1.1, BV421-anti-TCRβ, PE-anti-2B4, FITC-anti-CD107a, PE-Cy7-anti-CD39, PE-anti-FasL, PE-Cy7-anti-CRTAM, PE-anti-NKG2D, PE-anti-NKG2A/C/E, FITC- or AlexaFluor647-anti-GzB, PE-anti-Eomes, PE- or A647-anti-T-bet, PE-anti-FoxP3, AlexaFluor647-anti-Blimp1, and PE-anti-PRF were used. All mAbs were purchased from Biolegend, BD Biosciences or eBioscience/Thermo Fisher Scientific. Spleen cells were stained after red blood cell lysis with ACK buffer. Hearts were perfused with 20 mL of cold PBS, minced into small fragments and treated with collagenase D (1.0 mg/mL) and DNAse I (0.1 mg/mL), for 1 hr at 37°C. Collagenase D, DNAse I, monensin and saponin were purchased from Sigma-Aldrich. Single cell suspensions were incubated with anti-CD16/CD32 (FcR block) for 5 min and then surface stained for 30 min on ice. For IL-10 and IFN-γ intracellular staining, 2 × 10[6] spleen cells were cultured in the presence of PMA (16 nM), ionomycin (0.5 μM), monensin (5 μM) and brefeldin A (3.0 μg/mL) for 4 hr. After staining for surface markers, cells were fixed with paraformaldehyde 1% for 1 hr, permeabilized with saponin 0.2% for 20 min and stained with PE-Cy7-anti-IFN-γ mAb and/or PE-anti-IL-10 mAbs (Biolegend). For TF and GzB/PRF staining,

the FoxP3 staining buffer kit (eBioscience, ThermoFisher Scientific) was employed. At least 20,000 events gated on CD4+ T cells were acquired. Analytical flow cytometry was conducted with MoFlo (Beckman Coulter/Dako-Cytomation), FacsCanto, FacsCalibur or BD LSRFortessa (BD Bioscience) and data analyzed with Summit V4.3 software (Dako Colorado, Inc).

## Cytotoxicity assays and cell lines

For redirected cytotoxic assay, sorted B cells (B220+), previously activated by LPS (InvivoGen), at 25 µg/mL for 48 hr and then incubated for 1 hr with anti-CD3ε antibody (10 µg/mL), were employed as target cells. FACS-sorted naïve (CD44$^{low}$CD4+) and effector/memory (CD44$^{hi}$CD4+) T cells were purified from the spleen of non-infected and infected mice, respectively. In some experiments, sorted T cells were treated for 1 h with GzB (Z-AAD-CMK, 10 µM) and PRF (Concamycin A, 0.5 µM) inhibitors (Biovision Inc, Milpitas, CA), before coincubation at different E:T ratios, with $10^4$ target B cells (prepared as aforementioned) in round-bottom microplates (Costar-Corning) for 14 hr, at 37°C in a 5% $CO_2$ incubator and in triplicates. After incubation, living target cells were quantified by flow cytometry, after staining with propidium iodide at 0.5 µg/mL (Sigma-Aldrich). For *T. cruzi*-specific cytotoxicity assays, CD4+ T cells were obtained from infected B6 mice, at day 14 pi, and purified by FACS sorting (MoFlo cell sorter, Beckman Coulter/Dako-Cytomation). IC-21 macrophage cell lineage (ATCC) were used as targets in vitro. For this, IC-21 cells were counted and split into two tubes: half of the target cells were incubated for 45 min at 37° C in 5% $CO_2$, in RPMI containing *T. cruzi* amastigote protein extract (30 µg/mL) obtained as previously described (*Oliveira et al., 2010*). For in vivo cytotoxic assays, Ag-loaded cells were then stained with 2.5 µM CFSE (CFSE$^{Hi}$) (Invitrogen, Life Technologies), and control cells with 0.5 µM CFSE (CFSE$^{Low}$) in 5 mL of warm PBS for 10 min at 37 ° C. After two washes in RPMI 1640 (10% FCS), cells were resuspended at a concentration of $2 \times 10^5$ cells/mL each and mixed at 1:1 ratio (CFSE$^{Hi}$:CFSE$^{Low}$). Purified CD4+ T cells were added at different E:T ratios. Cells were seeded in duplicate to round-bottom 96-well plate in a final volume of 200 µL. After ON incubation, cells were harvested, incubated with 0.5 µg/mL of propidium iodide and PI+ cells were excluded from FACS analysis. The percentage of specific lysis was calculated using the formula: % of specific lysis $=100\times [(CFSE^{Low} / CFSE^{Hi})$ ctr $- (CFSE^{Low} / CFSE^{Hi})$ effector] $\times (CFSE^{Low} / CFSE^{Hi})$ ctr$^{-1}$. For in vivo cytotoxicity assays, splenocytes from non-infected B6 mice were employed as target cells after being divided in 3 subpopulations and stained with 10.0 (High), 2.5 (Int), and 0.625 (Low) µM of CFSE or of CTV (Invitrogen, Life Technologies). CFSE$^{Int}$ and CFSE$^{Low}$ cells were loaded with one of the following peptides: SA85c (SHNFTLVASVIIEEA), SA85d (LVASVIIEEAPSGNT), OVA$_{323-339}$, OVA$_{257-263}$, or PA8 (VNHRFTLV) at 2 µM, all from GenScript, EUA. CFSE$^{Hi}$ cells were not loaded with any peptide and used as internal control. For calculating the percentage of specific lysis, CFSE$^{Hi}$ and CFSE$^{Int}$ (or CFSE$^{Low}$) were gated together in order to constitute 100% of cells. CFSE$^{Hi}$ and CFSE$^{Int}$ (or CFSE$^{Low}$) populations were then individually gated and specific lysis was calculated using the following formula: % of specific lysis $=100\times [(CFSE^{Low\ or\ Int} / CFSE^{Hi})$ ctr $- (CFSE^{Low\ or\ Int} / CFSE^{Hi})$ effector] $\times (CFSE^{Low\ or\ Int} / CFSE^{Hi})$ ctr$^{-1}$.

## Real time RT-PCR

RNA was extracted using the RNeasykit (QIAGEN). cDNA was synthesized using the High Capacity cDNA Reverse Transcription Kit (Invitrogen, Life Technologies), following the manufacturer's instructions. Gene expression was analyzed using primers in combination with SYBR Green on a StepOne-Real Time PCR System (Applied Biosystems, Life Technologies). The expression level of the genes of interest was determined relative to the expression of 18 S rRNA. Primers: *Runx3* Fwd.: 5'- ACA GCA TCT TTG ACT CCT TCC –3'; *Runx3* Rev.: 5- GTT GGT GAA CAC GGT GAT TG- 3'; *Thpok* Fwd: 5'- ATG GGA TTC AA TCA GGT CA-3'; *Thpok* Rev.5'- TTCTTC CTA CAC CCT GTG CC –3'; *CD8a* Fwd: 5' - ACT GCA AGG AAG CAA GTG GT – 3'; *CD8a* Rev: 5'-CAC CGC TAA AGG CAG TTC TC-3'; 18S-Fwd.: 5'- GGG CGT GGG GCG GAG ATA TGC - 3' and 18S-Rev.: 5'- CG CGG ACA CGA AGG CCC CAA A –3'. For parasite load quantification, hearts of infected mice were perfused, minced and immediately homogenized in 1.0 mL of 4.0 M Guanidine thiocyanate (SIGMA-Aldrich) containing 8.0 µl/mL of β-mercaptoethanol and processed for DNA extraction. Generation of PCR standards and detection of parasite tissue load by real-time PCR was carried out as described (*Oliveira et al., 2017*); briefly, primers amplify a repeated satellite sequence of *T. cruzi* DNA of 195 base-pairs: TCZ-Fwd.: (5'-GCTCTTGCCCACAAGGGTGC-3') and TCZ-Rev.: (5'-CCAAGCAGCGGATAGTTCAGG-3').

Reactions with TNF-a-Fwd: (5'- CCTGGAGGAGAAGAGGAAAGAGA-3') and TNF-a-Rev.: (5'- TTGA GGACCTCTGTGTATTTGTCAA-3') primers for *Mus musculus* TNF-α gene were used as loading controls.

## Gene-expression analysis

Gene expression data were extracted with GeneSpring software GX 11.0 software using default parameters. Data were then normalized by subtracting the average from all genes from each gene intensity and dividing the result by the standard deviation of all gene intensities (i.e. z-score normalization). Z ratios were calculated as previously described (*Cheadle et al., 2003*) and used to compare the difference in expression between WT samples and *Myd88*-/- samples. Gene Set Enrichment Analysis (GSEA) (http://www.broadinstitute.org/gsea/index.jsp) was performed on the genes pre-ranked by Z ratios (1,000 permutations and Weighted Enrichment Statistic) using as data set 82 differentially expressed genes of sorted WT and *Myd88*-/- CD4+ T cells (GEO accession no. GSE57738) (*Oliveira et al., 2017*); and using as gene set the CD4+GzmB+ (CD4CTL) transcriptional signature (159 upregulated genes) previously described (https://www.ebi.ac.uk/ena/data/view/PRJEB14043).

## Adoptive CD4+ T cell transfer assay

Eight-week-old male GzmbCreER$^{T2}$/ROSA26EYFP mice were infected with $2 \times 10^3$ trypomastigotes and treated with 4 mg/animal/day of tamoxifen citrate (Sandoz-Novartis) from day 7 to 11 pi by gavage. Spleen cells were harvested and treated with ACK for red blood cells lysis at day 13 pi. Alternatively, CD4+ T cells were purified from the spleen of non-treated and infected *Ifng*-/- mice. In both cases, spleen cells were labeled with biotinylated anti-B220, anti-CD11b, anti-NK1.1, anti-γδTCR, anti-CD8 and PE-conjugated anti-CD4 antibodies for 30 min on ice, followed by staining with PE-Cy7-conjugated Streptavidin for 20 min. Residual PE-Cy7+ cells were excluded and EYFP+PE+ (CD4+GzB+ T cells) or PE+CD4+ (*Ifng*-/- CD4+ T cells) were sorted using MoFlo cell sorter (Beckman Coulter/Dako-Cytomation). CD4+GzB+ T cells ($2.8 \times 10^5$ cells/animal) or *Ifng*-/- CD4+ T cells ($1.3 \times 10^6$ cells/animal), both presenting purity higher than 98%, were adoptively transferred by iv injection into *Il18ra*-/- recipient mice, previously infected with $1 \times 10^3$ trypomastigotes (day 7 pi). Control groups of *Il18ra*-/- and *Myd88*-/- infected mice received PBS. Survival was accompanied daily.

## Statistics

Statistical analyses were performed using GraphPad Prism version 6 for Mac OS (GraphPad Software, San Diego California USA, https://www.graphpad.com). Sample sizes were chosen to allow the detection of statistical differences with $p < 0.5$ using Student's t test, or the Log-rank (Mantel-Cox) test for mouse survival curves. Correlation analysis employed Pearson correlation test. When comparing two groups, the minimum number was n=3 per group. In some experiments, the number was increased to n=9 to permit $p < 0.05$ even considering the possibility of a single outlier in a population. Data are expressed as mean ± SEM and considered statistically significant if p values were<0.05.

## Acknowledgements

We would like to thank Dr Ricardo T Gazzinelli (Fiocruz, MG, Brazil) for enriching discussion and providing GzmbCreER$^{T2}$/ROSA26EYFP mouse lineage, Dr Helder I Nakaya (University of São Paulo, SP, Brazil) for helping with Bioinformatics Analyses and Dr Hilde Cheroutre (La Jolla Institute for Immunology, CA, USA) for critically reading our manuscript and enriching discussion. Funding: this study was financed in part by the Coordenação de Aperfeiçoamento de Pessoal de Nível Superior - Brasil (CAPES) - Finance Code 001; Fundação Carlos Chagas Filho de Amparo à Pesquisa do Estado do Rio de Janeiro (FAPERJ); Conselho Nacional de Pesquisas (CNPq) and by The National Institute for Vaccine Development and Technology (INCTV-CNPq). CDB and LMB received PhD fellowships from CNPq; AG and EGAN received PhD fellowships from CAPES; FBC and AG received PDJ-CNPq fellowships. GM received PIBIC-CNPq fellowship. AN, WOD and MB received PQ fellowship from CNPq; AN and MB received CNE grant from FAPERJ.

The funders had no role in study design, data collection and analysis, decision to publish or preparation of the manuscript.

## Additional information

### Funding

| Funder | Grant reference number | Author |
|---|---|---|
| Fundação Carlos Chagas Filho de Amparo à Pesquisa do Estado do Rio de Janeiro | CNE E-26/202.603/2019 | Maria Bellio |
| Conselho Nacional de Desenvolvimento Científico e Tecnológico | PQ Award 312143/2018-4 | Maria Bellio |
| Conselho Nacional de Desenvolvimento Científico e Tecnológico | INCT -V 465293/2014-0 | Maria Bellio |
| Coordenação de Aperfeiçoamento de Pessoal de Nível Superior | Finance Code 001 | Maria Bellio |
| Fundação Carlos Chagas Filho de Amparo à Pesquisa do Estado do Rio de Janeiro | CNE E26/202.942/2016 | Alberto Nóbrega |
| Fundação de Amparo à Pesquisa do Estado de Minas Gerais | Universal 2018 | Walderez O Dutra |
| Fundação Carlos Chagas Filho de Amparo à Pesquisa do Estado do Rio de Janeiro | APQ1 Manutenção da Unidade de Citometria. E26/210.986/2016 | Alberto Nóbrega |

The funders had no role in study design, data collection and interpretation, or the decision to submit the work for publication.

### Author contributions

Carlos-Henrique D Barbosa, Ariel Gomes, Formal analysis, Investigation, Methodology; Fábio B Canto, Alberto Nóbrega, Formal analysis, Investigation, Methodology, Writing – review and editing; Layza M Brandao, Ana-Carolina Oliveira, Methodology, Formal analysis, Investigation; Jéssica R Lima, Methodology, Investigation; Guilherme A Melo, Eula GA Neves, Investigation, Methodology, Formal analysis; Alessandra Granato, Methodology; Walderez O Dutra, Data curation, Formal analysis, Investigation; Maria Bellio, Conceptualization, Data curation, Formal analysis, Funding acquisition, Investigation, Methodology, Project administration, Validation, Writing – original draft, Writing – review and editing, Supervision

### Author ORCIDs

Carlos-Henrique D Barbosa ⓘ http://orcid.org/0000-0001-7644-7034
Fábio B Canto ⓘ http://orcid.org/0000-0001-7574-2613
Walderez O Dutra ⓘ http://orcid.org/0000-0002-7586-9996
Ana-Carolina Oliveira ⓘ http://orcid.org/0000-0002-0036-3720
Alberto Nóbrega ⓘ http://orcid.org/0000-0002-9840-0103
Maria Bellio ⓘ http://orcid.org/0000-0002-3360-2740

### Ethics

Human subjects: All individuals included in this work were volunteers and treatment and clinical care were offered to all patients, as needed, despite their enrollment in this research project. All participants provided written informed consent for participation in the study. This cross-sectional study is part of an extended project evaluating biomarkers of cardiomyopathy development in Chagas disease, which has the approval of the National Committee of Ethics in Research (CONEP#195/2007) and are in accordance with the Declaration for Helsinki.

All mouse experiments were conducted in accordance with guidelines of the Animal Care and Use Committee of the Federal University of Rio de Janeiro (Comitê de Ética do Centro de Ciências da Saúde CEUA - CCS/UFRJ, license: IMPPG022).

### Decision letter and Author response
Decision letter https://doi.org/10.7554/eLife.74636.sa1
Author response https://doi.org/10.7554/eLife.74636.sa2

---

## Additional files

### Supplementary files
• Transparent reporting form

### Data availability
Transcriptome data was uploaded to GEO and is already publicly available with the GEO accession no. GSE57738.

The following previously published datasets were used:

| Author(s) | Year | Dataset title | Dataset URL | Database and Identifier |
|---|---|---|---|---|
| Reis BS, Bellio M, Nakaya HI, Nóbrega A | 2014 | T cell-intrinsic MyD88 signaling in cognate immune response to intracellular parasite infection: crucial role for IL-18R | https://www.ncbi.nlm.nih.gov/geo/query/acc.cgi?acc=GSE57738 | NCBI Gene Expression Omnibus, GSE57738 |
| Donnarumma T | 2016 | Single-cell RNA sequencing of CD4 T cells 7 days after infection or immunisation | https://www.ebi.ac.uk/ena/browser/view/PRJEB14043 | ENA, PRJEB14043 |

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
