## [Editor Report]

This is an interesting study, conducted in mice, that demonstrates for the first time the presence of a large population of cytotoxic CD4^+^ T lymphocytes in infection with Trypanosoma cruzi, a relevant human pathogen. At present, the relevance of these cells in protective immunity engendered by the host remains unclear. Additional experiments are needed to characterize the functionality of these cytotoxic CD4 T cells vis-a-vis the canonical Th1 T cells. This paper can be of interest to scientists interested in immune responses to parasitic infections.

---

## [Decision Letter]

**Decision letter after peer review:**

Thank you for submitting your article "Cytotoxic CD4^+^ T cells driven by T-cell intrinsic IL-18R/MyD88 signaling predominantly infiltrate *Trypanosoma cruzi*-infected hearts" for consideration by *eLife*. Your article has been reviewed by 3 peer reviewers, and the evaluation has been overseen by a Reviewing Editor and Satyajit Rath as the Senior Editor. The following individual involved in review of your submission has agreed to reveal their identity: Tiziano Donnarumma (Reviewer #3).

Essential revisions:

1) The authors are encouraged to perform additional experiments in support of their claims:

(a) whether CD4 CTL cells are protective or pathological and

(b) how is their function is coordinated with or distinct from the role of "classical" Th1 cells that the authors previously showed play and essential role.

2) These experiments are necessary to establish "functional" and/or "protective" attributes of the CD4T CTL cells to determine if their killing capability is dependent on the phenotype described, namely Gzb-mediated killing.

3) A comparison over the state of activation/exhaustion of heart and lymphoid CD4 CTL cells will complement and link with the data on protection and functionality.

*Reviewer #1 (Recommendations for the authors):*

– The y-axes on most of the bar graphs are rather complicated and duplicate information provided on the x-axes (e.g. Fig. 1F)

– Fig. 3 F-H is poorly explained. At best, 10% GzB+CD4+ T cells are positive for NKG2, NKGD2 or CRTAM. It is not clear whether the same subpopulation expresses all 3 markers (although in hearts at least the latter two are co-expressed on GzB+ CD4+ T lymphocytes).

*Reviewer #2 (Recommendations for the authors):*

It would be essential, in my view, for the authors to separate FasL-mediated from granzyme/perforin-mediated cytotoxicity (e.g. the use of granzyme substrates loaded on target cells or of perforin-deficient CD4 cells).

If IL-18R or MyD88 deficient CD4 cells, which clearly lack CD4CTL development and protective function, still express FasL and can kill in a FasL-dependent manner, perhaps this can be used to argue that FasL expression is not sufficient for protection in vivo by CD4CTLs.

In light of the authors' previous finding that the IL-18R/MyD88 signaling pathway is important for development of protective Th1 responses, the assertion that "It is expected that pharmacological targeting of the IL-18R/MyD88 signaling pathway might contribute to the strategic development of new therapeutic and vaccine approaches aiming the control of the CD4CTL response", may need to be revised. Can such treatments be CD4CTL-specific and spare Th1 cells?

Also, the statement that correlations between levels of CD4CTLs and measures of heart dysfunction in chronic Chagas cardiomyopathy "suggest that CD4CTLs may play a pathogenic role in the chronic phase of Chagas disease" may need to be qualified, as the alternative possibility that both of these simply reflect parasite burden in the heart, without a causal link between them, would have to be entertained, if not excluded.

*Reviewer #3 (Recommendations for the authors):*

– On Figure 1G the % of CD4 GzB+ is much lower than what shown in Figure 1B,D.

– The cytoxicity experiments are very convincing. Using GzB YFP+ CD4 could further demonstrate the superior cytolytic capability of the CDCTLs subset.

– FoxP3 does not correlate with GzB production in CD4 in the model and interestingly seems to be decreased in infected animal. Could the author comment on the possibility that lower Treg numbers could be favourable to CD4 CTLs development?

– In Fig2H the authors compare levels of ThPOK and RUNX3 in CD4 GzB vs naive cells showing a noticeable RunX3 upregulation in CD4CTLs. Instead of naive cells CD44H^+^ GzB- cells (or the total CD4 population) should be used for the comparison.

– TCF7gene /TCF1 protein has been linked to CD4 and CD8 stemness and his loss is associated to a a terminally differentiated state. Loss of TCF1 is also the most recognizable feature of the CD4 CTLs gene signature used as a reference in the paper. Did the authors checked whether TCF1 levels are diminished in CD4CTLs in their model?

– The amount of CD4 CTLs found in the heart and the massive coexpression of NKG2D and CRTAM are striking and very promising in the view of future studies aimed at establishing the role of CD4CTLs in the pathology.

---

## [Author Response]

Essential revisions:1) The authors are encouraged to perform additional experiments in support of their claims:(a) whether CD4 CTL cells are protective or pathological and(b) how is their function is coordinated with or distinct from the role of "classical" Th1 cells that the authors previously showed play and essential role.

Our results in the mouse model of infection with *T. cruzi*, employing the adoptive transfer of WT CD4^+^GzB+ T cells to the susceptible *IL18ra^-/-^* mouse strain, clearly indicate a beneficial role of transferred CD4CTLs. Although the adoptive transfer of CD4^+^GzB+ T cells did not decrease the parasite load (Figure 6G, right panel), it resulted in a significant extent of survival of the transferred group (Figure 6G, left panel). Therefore, overall, CD4CTLs are exerting a protective effect in infected mice. Please see more details about this topic in our answer to Reviewer #1. In order to avoid possible misunderstanding, we changed “protective” for “beneficial” in our text. These data are evidence that the classical IFN-greleasing function of Th1, which has known microbicide action, is not being exerted by transferred CD4^+^GzB+ T cells. This result is in accordance to our finding that CD4^+^GzB+ T cells are for the great majority non-producers of IFN-g (Figure 1F). We also showed that *IFNγ^-/-^* mice have the same percentage of GzB+ cells among CD4^+^ T lymphocytes compared to WT mice (Figure 4B) and that, similarly, the adoptive transfer of *IFNγ^-/-^* CD4^+^ T cells significantly extend survival of susceptible mice (Figure 6Supp.Figure 2), albeit it does not diminish parasite levels (Oliveira *et al.*, *eLife*, 2017). Therefore, these data strongly suggest that the transferred-CD4^+^GzB+ T cells are extending survival of recipient mice, independently of classic Th1 function. It would be interesting to have a GzB-reporter mice lacking the *IFNγ* gene in CD4 T cells only, in order to obtain purified CD4^+^GzB+IFNγ-negative cells for adoptive transfer experiments. Unfortunately, we are not aware of the existence of such a mouse strain. As discussed in the manuscript, together with our previous work, the current results point to complementary roles exerted by CD4CTL and Th1 cells. Both cell types might be beneficial against infection with *T. cruzi*, working together to establish a balance between the Th1-dependent control of parasite load on one side, and CD4CTLmediated dampening of the immunopathology resulting from this process, on the other. Please see our revised discussion on the function of CD4CTLs and Th1 cells on pages 20-21 of the revised manuscript.

Concerning a possible pathological function for CD4CTLs: The correlation found between the percentage of circulating PRF+CD4^+^ T cells and the severity of chronic myocarditis in patients with Chagas Disease raises the possibility of a detrimental role of this cell subset during the chronic phase of the infection. Clearly, this correlation does not prove that CD4CTLs are directly involved in the immunopathology, although at present this possibility cannot be excluded. Although speculative, this hypothesis deserves discussion. Thus, we improved the discussion on this point, please see pages 18 and 19 (lines 445-448) and page 22 (lines 535-539) of the revised manuscript. Assessing the exact role of this human cytotoxic CD4^+^ T subpopulation in Chagas disease requires further investigation. Of note, in humans, myocarditis is also uniformly present during acute infection, although difficult to diagnose (Henao-Martínez et al., 2012, Trans R Soc Trop Med Hyg 106(9):521). The mechanism linking the acute to the chronic myocardial process, however, are not known yet. Please also note that a beneficial or pathological function of any T cell subset can vary in different phases (acute x chronic) of a disease. In fact, many groups, including ours, have shown the protective role of Th1 cells in the acute phase of infection with *T. cruzi*, while, on the other hand, the Th1 response has also been associated with a pathogenic role in Chagas chronic myocarditis (Oliveira et al., 2017, and reviewed in Ferreira et al., 2014 World J Cardiol 2014 6(8): 7820 and in Fresno and Girones, 2018, Front.Immunol. 9;351). Hypothetically, extensive tissue damage might be a consequence of a failure in the regulatory circuits, which would otherwise control effector T cell subsets, avoiding pathology. The persistence of a long lasting response in infected tissues, without adequate immunoregulation, could lead to the pathogenesis observed in the chronic phase of Chagas Disease.

2) These experiments are necessary to establish "functional" and/or "protective" attributes of the CD4T CTL cells to determine if their killing capability is dependent on the phenotype described, namely Gzb-mediated killing.

In order to determine if CD4CTL killing capability is dependent on GzB and PRF mediated killing, we have now performed the cytotoxic assays in the presence of GzB and PRF inhibitors, (Z-AAD-CMK and Concamycin A). As shown in Figure 2 – Figure supp. 1A, the killing of target cells was totally inhibited in this condition. Therefore, the new data added to the revised manuscript demonstrate that the killing capability of the CD4CTL subset in *T. cruzi-*infection model is dependent on GzB- and PRFmediated action, as previously shown for tumor-specific CD4CTLs (Quezada et al., 2010).

3) A comparison over the state of activation/exhaustion of heart and lymphoid CD4 CTL cells will complement and link with the data on protection and functionality.

In our present study we shown that the great majority of CD4 T cells with cytotoxic phenotype expresses high levels of the activation/memory marker CD44 in the spleen (Figure 2—figure supplement 2). We now added to the revised version of the manuscript the information that the totality of CD4^+^ T cells in the heart express the CD44 marker (Figure 7—figure supplement 2). We also added new cytometry data showing the frequencies of CD4CTLs (GzB+PRF+) that express the activation/exhaustion markers PD-1, Tim-3, Lag-3 and CD39, and their corresponding MFI, in the spleen and in the heart, compared to other activated CD4^+^T cells (CD44^hi^) expressing neither GzB nor PRF (Figure 7—figure supplement 3). Furthermore, the comparison of the level of expression of these markers on CD4CTLs and on CD8CTLs, as well as the frequency of activation/exhaustion marker-positive cells among these subsets in the spleen and in the heart of infected mice were also added to the revised manuscript (Figure 7—figure supplement 4). Please note that this point is further commented in the detailed response to Reviewer # 1, on pages 6-8, below.

We believe that, taken together, the new data and discussion included in the revised manuscript address to the Reviewers’ concerns and add for the better comprehension of CD4CTL biology in the model of acute infection with *T. cruzi*, which is being described for the first time by our work.

Reviewer #1 (Recommendations for the authors):– The y-axes on most of the bar graphs are rather complicated and duplicate information provided on the x-axes (e.g. Fig. 1F)

Thank you for the observation. Duplicated information on Fig.1 F graph axes was corrected.

– Fig. 3 F-H is poorly explained. At best, 10% GzB+CD4+ T cells are positive for NKG2, NKGD2 or CRTAM. It is not clear whether the same subpopulation expresses all 3 markers (although in hearts at least the latter two are co-expressed on GzB+ CD4+ T lymphocytes).

With the scope of obtaining brighter staining and due to limitations of our flow cytometry facility at the time, we employed PE-linked mAbs for both NKG2D and NKG2A/C/E staining, which were thus used in different panels. For CRTAM staining, a PE-Cy7 mAb was employed, allowing the analysis of double positive CRTAM+NKG2D+ and CRTAM+NKG2A/C/E+ CD4+ T cells. Our results (Figure 7E and Figure 7-figure supplement 1D) clearly show that in the heart the vast majority of GzB+CD4+ T cells express both CRTAM and NKG2D (but not NKG2A/C/E). We have now analyzed the co-expression of CRTAM and NKG2D or CRTAM and NKG2A/C/E in the GzB+CD4+ T cells of the spleen and these data were added to new Figure 3-figure supplement 3. We found that while half of the CRTAM+ cells among CD4+GzB+ T splenocytes also express NKG2D, only a minority of CD4+GzB+ T cells co-express CRTAM and NKG2A/C/E. It is true that, in the spleen, the expression of these markers is limited to a lower frequency of cells, although both NKG2A/C/E+ and NKG2D+ cells are present at a similar ratio among CD4+GzB+ T cells. These data indicate that only one of the subpopulations present in the spleen, that is, double-positive CRTAM+NKG2D+ GzB+CD4+ T cells, migrate and/or are retained as infiltrating cells in the infected cardiac tissue. The difference observed on the expression of NKG2A/C/E or NKG2D by CD4CTLs infiltrating IAV infected lungs (Marshall et al., 2017, JI) or parasite infected hearts (our ms), respectively, is discussed on page 19, lines 457-464.

Reviewer #2 (Recommendations for the authors):It would be essential, in my view, for the authors to separate FasL-mediated from granzyme/perforin-mediated cytotoxicity (e.g. the use of granzyme substrates loaded on target cells or of perforin-deficient CD4 cells).

Previous studies in viral infection models have investigated the killing mechanism of CD4CTLs and have established that this population can employ both FasL/Fas- and Perforin/GzB-mediated killing mechanisms (reviewed in Brown, 2010). Of note, FasL/Fas-mediated cytotoxicity is not incompatible with the simultaneous action of Perforin and GzB in the target-cell killing process. In fact, it was shown that the CTL’s ‘kiss of death’ stems from the complementary action of FasL- and Perforin- mediated lytic mechanisms (Hassin et al., 2011, Immunology 133(2):190). In that study, the authors suggested that the complementary FasL/Fas pathway would be particularly important in cells with lower expression of Perforin/GzB, as it is known to be the case of CD4^+^GzB+ T cells, when compared to CD8CTLs. We have now addressed this question by performing the cytotoxic assay in the presence of GzB and PRF inhibitors, (Z-AAD-CMK and Concamycin A, respectively). As shown in new Figure 2 – Figure supp. 1A, the killing of target cells was totally abolished in the presence of these inhibitors, as previously shown for tumor-specific CD4CTLs (Quezada et al., 2010). Therefore, the new data added to the revised manuscript demonstrate that the lytic capability of the CD4CTL subset differentiating in the murine *T. cruzi-*infection model is dependent on GzB- and PRF-mediated action.

If IL-18R or MyD88 deficient CD4 cells, which clearly lack CD4CTL development and protective function, still express FasL and can kill in a FasL-dependent manner, perhaps this can be used to argue that FasL expression is not sufficient for protection in vivo by CD4CTLs.

Whether IL-18R- or MyD88-deficient CD4^+^ T cells express FasL or not is an interesting question. Our study shows that, in the absence of T-cell intrinsic IL18R/MyD88 signaling, fewer CD4CTLs are found in infected mice, probably reflecting the lower expansion of these cells (or of their precursor) as a consequence of decreased transcription levels of cell-cycle and anti-apoptotic genes in MyD88-/- CD4^+^ T cells, as reported in our previous work (Oliveira et al., 2017). In the present study, however, we did not focus on the killing capability of the few CD4CTLs present in IL18R- and MyD88-deficient mice. Nonetheless, the new results employing GzB and PRF inhibitors (please see Figure 2 – Figure supp. 1A and answer above), indicate that Fas/FasL contribution to the killing of target cells in our system occurs in coordination with PRF/GzB release at the cytotoxic synapsis, as previously described (Hassin et al., 2011, Immunology 133(2):190). Further investigation of the details on how the lack of IL-18R/MyD88 signaling can affect CD4CTL activity, including the expression of the FasL molecule, is undoubtedly an interesting question, but it cannot be addressed in a suitable time at the present.

In light of the authors' previous finding that the IL-18R/MyD88 signaling pathway is important for development of protective Th1 responses, the assertion that "It is expected that pharmacological targeting of the IL-18R/MyD88 signaling pathway might contribute to the strategic development of new therapeutic and vaccine approaches aiming the control of the CD4CTL response", may need to be revised. Can such treatments be CD4CTL-specific and spare Th1 cells?

Together, our present and past results (Oliveira et al., 2017) show that the T-cell intrinsic IL-18R/MyD88 signaling pathway is important for the development of both Th1 and CD4CTL responses. While our data indicate that CD4CTLs, as well as Th1 cells, are beneficial in the acute phase of infection with *T. cruzi*, it might be that, at the chronic stage of infection, CD4CTLs could have a detrimental role, as suggested by our data in humans (Figure 7G-J), and as discussed above. At present, the knowledge about CD4CTL function in cancer, infection and autoimmune diseases is still very limited. In our statement, we were referring to a hypothetical situation in which the enhancement (or inhibition) of CD4CTLs activity could be highly beneficial, independently of being simultaneously affecting the Th1 response. We believe that discovering how to specifically target the differentiation of CD4CTL or Th1 cell responses would probably be of high medical interest. However, we agree with Reviewer #2 that this statement might lead to misunderstanding and, thus, it was removed from the text of the revised manuscript.

Also, the statement that correlations between levels of CD4CTLs and measures of heart dysfunction in chronic Chagas cardiomyopathy "suggest that CD4CTLs may play a pathogenic role in the chronic phase of Chagas disease" may need to be qualified, as the alternative possibility that both of these simply reflect parasite burden in the heart, without a causal link between them, would have to be entertained, if not excluded.

We would like to thank Reviewer #2 for this comment. We totally agree that the correlation between the levels of CD4CTLs in the circulation and the echocardiographic parameters employed for evaluating myocarditis in Chagas cardiomyopathy patients is not a prove of causal link. We have now better presented these results, citing the alternative hypothesis on pages 18 and 19 (lines 445-448). However, please note that although a link between parasite persistence and the pathogenesis of Chagas heart disease has been established, the quantitative correlation between parasite load in the heart and myocarditis has not been fully demonstrated yet (reviewed in Lewis and Kelly, 2016, Trends in Parasitol.32(11):899). On the other side, although still debated, several studies have provided evidence for a role of autoimmunity in the clinical progression of chronic Chagas cardiomyopathy (reviewed in De Bona, 2018, Front Immunol 6;9:1842). Regardless of the mechanisms involved in the development of Chagas myocarditis, we agree that the higher levels of CD4CTLs found in Chagas patients might reflect not only elevated parasite burden but also inflammation in the heart, without necessarily implying a direct involvement of this cell subset in the pathology. Please see the revised discussion of this point on page 22 (lines 537-539).

Reviewer #3 (Recommendations for the authors):– On Figure 1G the % of CD4 GzB+ is much lower than what shown in Figure 1B,D.

It is common to have a certain variation in the frequency of GzB+CD4^+^ T cells in the spleen in different experiments, within the approximate range of 10 to 40% at day 14 pi. The kinetic experiments shown in Figure 1G were performed in the beginning of our study, years ago, when we were employing an anti-GzB mAb conjugated to FITC. After changing to A647-conjugated mAb, the percentage of detected GzB+ positive cells increased. Thereafter, we have been using the A647-conjugated mAb for detecting GzB, obtaining better results (at least 25% in the spleen at day 14 pi). With the exception of data shown in Figure 1G, all the other experiments showed in the manuscript were performed with an anti-GzB A647-conjugated mAb.

– The cytoxicity experiments are very convincing. Using GzB YFP+ CD4 could further demonstrate the superior cytolytic capability of the CDCTLs subset.

As explained above and in our answer to Reviewer #2, it is not possible to sort bona-fide GzB-negative cells to use as controls in a comparison experiment between GzB+ and GzB- CD4 T cells. However, our new results employing GzB and PRF inhibitors (Figure 2 – Figure supp. 1A), clearly show the dependency on these proteins for killing.

– FoxP3 does not correlate with GzB production in CD4 in the model and interestingly seems to be decreased in infected animal. Could the author comment on the possibility that lower Treg numbers could be favourable to CD4 CTLs development?

The role of Tregs in *T. cruzi* infection is still a controversial question. Some studies found that (i) Treg cells are not necessary for *T. cruzi* evasion of immune responses during acute or chronic infection; (ii) Tregs have limited role on host resistance; (iii) Treg cells do not play a major role in regulating CD8 T cell effector responses and pathogenesis (Kotner J. and Tarleton, R., 2007, IAI 75:861; Sales, PA. et al., 2008 Microbes Infect10(6):680). On the other hand, other works reported that (i) Treg depletion leads to reduced cardiac parasitism and inflammation accompanied by an augmented Th1 response; (ii) reduced Treg frequency inversely correlated with the magnitude of the effector immune response, as well as with emergence of acute immunopathology and the emergence of protective anti-parasite CD8^+^ T cell immunity (Bonney KM, 2015, Parasitol Res.114(3): 1167; Furlan, CLA, 2018, Front Immunol. 9: 2555.). Interestingly, in our study we also found that the ratio of spleen Tregs among CD4^+^ T cells decrease in mice infected with *T. cruzi* (Figure 2E). Therefore, it is possible that, indeed, the observed lower Treg ratios favor the CD4CTL response in T. cruzi-infected mice. However, for the moment, we have no data to address this question in more details and, thus, no information or insights for further comments on this subject.

– In Fig2H the authors compare levels of ThPOK and RUNX3 in CD4 GzB vs naive cells showing a noticeable RunX3 upregulation in CD4CTLs. Instead of naive cells CD44H^+^ GzB- cells (or the total CD4 population) should be used for the comparison.

In the experiment shown in Figure 2H, we employed total naïve CD4 T cells as negative control for the expression of Runx3. Besides the impossibility of obtaining purified bona fide GzB-negative cells from the mouse reporter strain (as explained above and in answers to Reviewer #2, an important consideration here is that Runx3 is also known to be expressed by Th1 cells (Djuretic et al., 2007). Therefore, since Runx3 expression is not an exclusive feature of CD4CTLs, some level of Runx3 mRNA would also be probably detected in the subset of CD44hi GzB-negative cells (which includes Th1 cells). However, an important result of the experiment shown in Figure 2H concerns the finding that CD4^+^ GzB+ T cells in the spleen do not downmodulate ThPOK, as opposed to what has been shown for IEL CD4CTLs in a non-infectious context (Mucida, D. et al., 2013, Nat Immunol). Please find the discussion on this point on page 23, lines 562-568 of the revised manuscript.

– TCF7gene /TCF1 protein has been linked to CD4 and CD8 stemness and his loss is associated to a a terminally differentiated state. Loss of TCF1 is also the most recognizable feature of the CD4 CTLs gene signature used as a reference in the paper. Did the authors checked whether TCF1 levels are diminished in CD4CTLs in their model?

Recent findings have redefined a role for T cell factor 1 (TCF1), which is not only involved in T cell development and T cell memory formation but also has functions in the differentiation pathways of effector CD4^+^ T lymphocytes. While Blimp1 suppresses Tcf7 expression, it is, in turn, negatively regulated by TCF-1. Accordingly, Donnarruma et al., 2016, have shown the loss of Tcf7 (TCF1) expression and the concomitant gain of Prdm1 (Blimp-1) expression in CD4CTLs developing in response to viral infection. In our present work, we show the higher frequency of cells expressing Blimp-1 among GzB+CD4^+^ T lymphocytes by flow cytometry (Figure 5D), as well as the higher expression of its coding gene, Prdm1, among WT compared to *Myd88^-/-^* CD4^+^ T cells by bioinformatics analysis of microarray data (Figure 5C). Therefore, one would expect to find lower levels of Tcf7 expression in WT CD4^+^ T cells, among which higher numbers of CD4CTLs are present, relative to *Myd88^-/-^* CD4^+^ T cells. On the other hand, however, *Tcf7* is associated with central T cell memory. We and others have shown the crucial role of T-cell intrinsic MyD88 signaling for the acquisition of memory cell signatures by CD4 T cells in response to infection or to immunization with Ag/adjuvant, respectively (Oliveira, 2017, *eLife* and Schenten, D. et al., 2014, Immunity 40:78). Two different specific probes for *Tcf7* were employed in the RNA microarray. For both probes, the analysis indicated a higher expression among Myd88 CD4^+^ T cells, but the fold change was lower than 1.5, the cut-off value we employed. Therefore, we did not include the result of the analysis of Tcf7 transcripts in RNA microarrays in the present manuscript. We believe that the result obtained reflects the fact that both CD4CTLs and memory CD4^+^ T cells are more abundant among WT than among *Myd88^-/-^* CD4^+^ T cells, but opposite results concerning the expression of Tcf7 in CD4CTLs and memory CD4^+^ T cell subsets are expected.

– The amount of CD4 CTLs found in the heart and the massive coexpression of NKG2D and CRTAM are striking and very promising in the view of future studies aimed at establishing the role of CD4CTLs in the pathology.

We thank Reviewer #3 for this comment. We agree that this can be very promising for understanding chronic Chagas myocarditis, as discussed on pages 22-23 (lines 541545) of the revised manuscript. We believe that investigating the interaction between CRTAM and/or NKG2D on T cells and their respective ligands expressed by the cardiac tissue might also bring novel insights for studies on other cardiac pathologies that involve infiltrating CD4^+^ T cells in this organ.